# Combinatorial Bandits Revisited

**Richard Combes**[*]    **M. Sadegh Talebi**[†]    **Alexandre Proutiere**[†]    **Marc Lelarge**[‡]

[*] Centrale-Supelec, L2S, Gif-sur-Yvette, FRANCE
[†] Department of Automatic Control, KTH, Stockholm, SWEDEN
[‡] INRIA & ENS, Paris, FRANCE

richard.combes@supelec.fr,{mstms,alepro}@kth.se,marc.lelarge@ens.fr

## Abstract

This paper investigates stochastic and adversarial combinatorial multi-armed bandit problems. In the stochastic setting under semi-bandit feedback, we derive a problem-specific regret lower bound, and discuss its scaling with the dimension of the decision space. We propose ESCB, an algorithm that efficiently exploits the structure of the problem and provide a finite-time analysis of its regret. ESCB has better performance guarantees than existing algorithms, and significantly outperforms these algorithms in practice. In the adversarial setting under bandit feedback, we propose COMBEXP, an algorithm with the same regret scaling as state-of-the-art algorithms, but with lower computational complexity for some combinatorial problems.

## 1  Introduction

Multi-Armed Bandit (MAB) problems [1] constitute the most fundamental sequential decision problems with an exploration vs. exploitation trade-off. In such problems, the decision maker selects an arm in each round, and observes a realization of the corresponding unknown reward distribution. Each decision is based on past decisions and observed rewards. The objective is to maximize the expected cumulative reward over some time horizon by balancing exploitation (arms with higher observed rewards should be selected often) and exploration (all arms should be explored to learn their average rewards). Equivalently, the performance of a decision rule or algorithm can be measured through its expected regret, defined as the gap between the expected reward achieved by the algorithm and that achieved by an oracle algorithm always selecting the best arm. MAB problems have found applications in many fields, including sequential clinical trials, communication systems, economics, see e.g. [2, 3].

In this paper, we investigate generic combinatorial MAB problems with linear rewards, as introduced in [4]. In each round $n \geq 1$, a decision maker selects an arm $M$ from a finite set $\mathcal{M} \subset \{0,1\}^d$ and receives a reward $M^\top X(n) = \sum_{i=1}^{d} M_i X_i(n)$. The reward vector $X(n) \in \mathbb{R}_+^d$ is unknown. We focus here on the case where all arms consist of the same number $m$ of *basic* actions in the sense that $\|M\|_1 = m, \ \forall M \in \mathcal{M}$. After selecting an arm $M$ in round $n$, the decision maker receives some feedback. We consider both (i) semi-bandit feedback under which after round $n$, for all $i \in \{1, \dots, d\}$, the component $X_i(n)$ of the reward vector is revealed if and only if $M_i = 1$; (ii) bandit feedback under which only the reward $M^\top X(n)$ is revealed. Based on the feedback received up to round $n-1$, the decision maker selects an arm for the next round $n$, and her objective is to maximize her cumulative reward over a given time horizon consisting of $T$ rounds. The challenge in these problems resides in the very large number of arms, i.e., in its combinatorial structure: the size of $\mathcal{M}$ could well grow as $d^m$. Fortunately, one may hope to exploit the problem structure to speed up the exploration of sub-optimal arms.

We consider two instances of combinatorial bandit problems, depending on how the sequence of reward vectors is generated. We first analyze the case of stochastic rewards, where for all

| Algorithm | LLR [9] | CUCB [10] | CUCB [11] | ESCB (Theorem 5) |
|---|---|---|---|---|
| **Regret** | $\mathcal{O}\left(\frac{m^3 d \Delta_{\max}}{\Delta_{\min}^2} \log(T)\right)$ | $\mathcal{O}\left(\frac{m^2 d}{\Delta_{\min}} \log(T)\right)$ | $\mathcal{O}\left(\frac{md}{\Delta_{\min}} \log(T)\right)$ | $\mathcal{O}\left(\frac{\sqrt{m}d}{\Delta_{\min}} \log(T)\right)$ |

Table 1: Regret upper bounds for stochastic combinatorial optimization under semi-bandit feedback.

$i \in \{1, \dots, d\}$, $(X_i(n))_{n \geq 1}$ are i.i.d. with Bernoulli distribution of unknown mean. The reward sequences are also independent across $i$. We then address the problem in the adversarial setting where the sequence of vectors $X(n)$ is arbitrary and selected by an *adversary* at the beginning of the experiment. In the stochastic setting, we provide sequential arm selection algorithms whose performance exceeds that of existing algorithms, whereas in the adversarial setting, we devise simple algorithms whose regret have the same scaling as that of state-of-the-art algorithms, but with lower computational complexity.

## 2 Contribution and Related Work

### 2.1 Stochastic combinatorial bandits under semi-bandit feedback

**Contribution.** (a) We derive an asymptotic (as the time horizon $T$ grows large) regret lower bound satisfied by any algorithm (Theorem 1). This lower bound is *problem-specific* and *tight*: there exists an algorithm that attains the bound on all problem instances, although the algorithm might be computationally expensive. To our knowledge, such lower bounds have not been proposed in the case of stochastic combinatorial bandits. The dependency in $m$ and $d$ of the lower bound is unfortunately not explicit. We further provide a simplified lower bound (Theorem 2) and derive its scaling in $(m, d)$ in specific examples.

(b) We propose ESCB (Efficient Sampling for Combinatorial Bandits), an algorithm whose regret scales at most as $\mathcal{O}(\sqrt{m} d \Delta_{\min}^{-1} \log(T))$ (Theorem 5), where $\Delta_{\min}$ denotes the expected reward difference between the best and the second-best arm. ESCB assigns an index to each arm. The index of given arm can be interpreted as performing likelihood tests with vanishing risk on its average reward. Our indexes are the natural extension of KL-UCB and UCB1 indexes defined for unstructured bandits [5, 21]. Numerical experiments for some specific combinatorial problems are presented in the supplementary material, and show that ESCB significantly outperforms existing algorithms.

**Related work.** Previous contributions on stochastic combinatorial bandits focused on specific combinatorial structures, e.g. $m$-sets [6], matroids [7], or permutations [8]. Generic combinatorial problems were investigated in [9, 10, 11, 12]. The proposed algorithms, LLR and CUCB are variants of the UCB algorithm, and their performance guarantees are presented in Table 1. Our algorithms improve over LLR and CUCB by a multiplicative factor of $\sqrt{m}$.

### 2.2 Adversarial combinatorial problems under bandit feedback

**Contribution.** We present algorithm COMBEXP, whose regret is $\mathcal{O}\left(\sqrt{m^3 T(d + m^{1/2}\underline{\lambda}^{-1}) \log \mu_{\min}^{-1}}\right)$, where $\mu_{\min} = \min_{i \in [d]} \frac{1}{m|\mathcal{M}|} \sum_{M \in \mathcal{M}} M_i$ and $\underline{\lambda}$ is the smallest nonzero eigenvalue of the matrix $\mathbb{E}[MM^\top]$ when $M$ is uniformly distributed over $\mathcal{M}$ (Theorem 6). For most problems of interest $m(d\underline{\lambda})^{-1} = \mathcal{O}(1)$ [4] and $\mu_{\min}^{-1} = \mathcal{O}(\text{poly}(d/m))$, so that COMBEXP has $\mathcal{O}(\sqrt{m^3 dT \log(d/m)})$ regret. A known regret lower bound is $\Omega(m\sqrt{dT})$ [13], so the regret gap between COMBEXP and this lower bound scales at most as $m^{1/2}$ up to a logarithmic factor.

**Related work.** Adversarial combinatorial bandits have been extensively investigated recently, see [13] and references therein. Some papers consider specific instances of these problems, e.g., shortest-path routing [14], $m$-sets [15], and permutations [16]. For generic combinatorial problems, known regret lower bounds scale as $\Omega\left(\sqrt{mdT}\right)$ and $\Omega\left(m\sqrt{dT}\right)$ (if $d \geq 2m$) in the case of semi-bandit and bandit feedback, respectively [13]. In the case of semi-bandit feedback, [13] proposes

| Algorithm | Regret |
|---|---|
| Lower Bound [13] | $\Omega\left(m\sqrt{dT}\right)$, if $d \geq 2m$ |
| COMBAND [4] | $\mathcal{O}\left(\sqrt{m^3 dT \log \frac{d}{m}\left(1 + \frac{2m}{d\underline{\lambda}}\right)}\right)$ |
| EXP2 WITH JOHN'S EXPLORATION [18] | $\mathcal{O}\left(\sqrt{m^3 dT \log \frac{d}{m}}\right)$ |
| COMBEXP (Theorem 6) | $\mathcal{O}\left(\sqrt{m^3 dT \left(1 + \frac{m^{1/2}}{d\underline{\lambda}}\right) \log \mu_{\min}^{-1}}\right)$ |

Table 2: Regret of various algorithms for adversarial combinatorial bandits with bandit feedback. Note that for most combinatorial classes of interests, $m(d\underline{\lambda})^{-1} = \mathcal{O}(1)$ and $\mu_{\min}^{-1} = \mathcal{O}(\mathrm{poly}(d/m))$.

OSMD, an algorithm whose regret upper bound matches the lower bound. [17] presents an algorithm with $\mathcal{O}(m\sqrt{dL_T^\star \log(d/m)})$ regret where $L_T^\star$ is the total reward of the best arm after $T$ rounds.

For problems with bandit feedback, [4] proposes COMBAND and derives a regret upper bound which depends on the structure of arm set $\mathcal{M}$. For most problems of interest, the regret under COMBAND is upper-bounded by $\mathcal{O}(\sqrt{m^3 dT \log(d/m)})$. [18] addresses generic linear optimization with bandit feedback and the proposed algorithm, referred to as EXP2 WITH JOHN'S EXPLORATION, has a regret scaling at most as $\mathcal{O}(\sqrt{m^3 dT \log(d/m)})$ in the case of combinatorial structure. As we show next, for many combinatorial structures of interest (e.g. $m$-sets, matchings, spanning trees), COMB-EXP yields the same regret as COMBAND and EXP2 WITH JOHN'S EXPLORATION, with lower computational complexity for a large class of problems. Table 2 summarises known regret bounds.

**Example 1: $m$-sets.** $\mathcal{M}$ is the set of all $d$-dimensional binary vectors with $m$ non-zero coordinates. We have $\mu_{\min} = \frac{m}{d}$ and $\underline{\lambda} = \frac{m(d-m)}{d(d-1)}$ (refer to the supplementary material for details). Hence when $m = o(d)$, the regret upper bound of COMBEXP becomes $\mathcal{O}(\sqrt{m^3 dT \log(d/m)})$, which is the same as that of COMBAND and EXP2 WITH JOHN'S EXPLORATION.

**Example 2: matchings.** The set of arms $\mathcal{M}$ is the set of perfect matchings in $\mathcal{K}_{m,m}$. $d = m^2$ and $|\mathcal{M}| = m!$. We have $\mu_{\min} = \frac{1}{m}$, and $\underline{\lambda} = \frac{1}{m-1}$. Hence the regret upper bound of COMBEXP is $\mathcal{O}(\sqrt{m^5 T \log(m)})$, the same as for COMBAND and EXP2 WITH JOHN'S EXPLORATION.

**Example 3: spanning trees.** $\mathcal{M}$ is the set of spanning trees in the complete graph $\mathcal{K}_N$. In this case, $d = \binom{N}{2}$, $m = N - 1$, and by Cayley's formula $\mathcal{M}$ has $N^{N-2}$ arms. $\log \mu_{\min}^{-1} \leq 2N$ for $N \geq 2$ and $\frac{m}{d\underline{\lambda}} < 7$ when $N \geq 6$, The regret upper bound of COMBAND and EXP2 WITH JOHN'S EXPLORATION becomes $\mathcal{O}(\sqrt{N^5 T \log(N)})$. As for COMBEXP, we get the same regret upper bound $\mathcal{O}(\sqrt{N^5 T \log(N)})$.

## 3 Models and Objectives

We consider MAB problems where each arm $M$ is a subset of $m$ *basic* actions taken from $[d] = \{1, \ldots, d\}$. For $i \in [d]$, $X_i(n)$ denotes the reward of basic action $i$ in round $n$. In the stochastic setting, for each $i$, the sequence of rewards $(X_i(n))_{n \geq 1}$ is i.i.d. with Bernoulli distribution with mean $\theta_i$. Rewards are assumed to be independent across actions. We denote by $\theta = (\theta_1, \ldots, \theta_d)^\top \in \Theta = [0,1]^d$ the vector of unknown expected rewards of the various basic actions. In the adversarial setting, the reward vector $X(n) = (X_1(n), \ldots, X_d(n))^\top \in [0,1]^d$ is arbitrary, and the sequence $(X(n), n \geq 1)$ is decided (but unknown) at the beginning of the experiment.

The set of arms $\mathcal{M}$ is an arbitrary subset of $\{0,1\}^d$, such that each of its elements $M$ has $m$ basic actions. Arm $M$ is identified with a binary column vector $(M_1, \ldots, M_d)^\top$, and we have $\|M\|_1 = m, \forall M \in \mathcal{M}$. At the beginning of each round $n$, a policy $\pi$, selects an arm $M^\pi(n) \in \mathcal{M}$ based on the arms chosen in previous rounds and their observed rewards. The reward of arm $M^\pi(n)$ selected in round $n$ is $\sum_{i \in [d]} M_i^\pi(n) X_i(n) = M^\pi(n)^\top X(n)$.

We consider both semi-bandit and bandit feedbacks. Under semi-bandit feedback and policy $\pi$, at the end of round $n$, the outcome of basic actions $X_i(n)$ for all $i \in M^\pi(n)$ are revealed to the decision maker, whereas under bandit feedback, $M^\pi(n)^\top X(n)$ only can be observed.

Let $\Pi$ be the set of all feasible policies. The objective is to identify a policy in $\Pi$ maximizing the cumulative expected reward over a finite time horizon $T$. The expectation is here taken with respect to possible randomness in the rewards (in the stochastic setting) and the possible randomization in the policy. Equivalently, we aim at designing a policy that minimizes regret, where the regret of policy $\pi \in \Pi$ is defined by:

$$R^\pi(T) = \max_{M \in \mathcal{M}} \mathbb{E}\left[\sum_{n=1}^T M^\top X(n)\right] - \mathbb{E}\left[\sum_{n=1}^T M^\pi(n)^\top X(n)\right].$$

Finally, for the stochastic setting, we denote by $\mu_M(\theta) = M^\top \theta$ the expected reward of arm $M$, and let $M^\star(\theta) \in \mathcal{M}$, or $M^\star$ for short, be any arm with maximum expected reward: $M^\star(\theta) \in \arg\max_{M \in \mathcal{M}} \mu_M(\theta)$. In what follows, to simplify the presentation, we assume that the optimal $M^\star$ is unique. We further define: $\mu^\star(\theta) = M^{\star\top}\theta$, $\Delta_{\min} = \min_{M \neq M^\star} \Delta_M$ where $\Delta_M = \mu^\star(\theta) - \mu_M(\theta)$, and $\Delta_{\max} = \max_M(\mu^\star(\theta) - \mu_M(\theta))$.

## 4 Stochastic Combinatorial Bandits under Semi-bandit Feedback

### 4.1 Regret Lower Bound

Given $\theta$, define the set of parameters that cannot be distinguished from $\theta$ when selecting action $M^\star(\theta)$, and for which arm $M^\star(\theta)$ is suboptimal:

$$B(\theta) = \{\lambda \in \Theta : M_i^\star(\theta)(\theta_i - \lambda_i) = 0, \, \forall i, \, \mu^\star(\lambda) > \mu^\star(\theta)\}.$$

We define $\mathcal{X} = (\mathbb{R}^+)^{|\mathcal{M}|}$ and $\mathrm{kl}(u,v)$ the Kullback-Leibler divergence between Bernoulli distributions of respective means $u$ and $v$, i.e., $\mathrm{kl}(u,v) = u\log(u/v) + (1-u)\log((1-u)/(1-v))$. Finally, for $(\theta, \lambda) \in \Theta^2$, we define the vector $\mathrm{kl}(\theta, \lambda) = (\mathrm{kl}(\theta_i, \lambda_i))_{i \in [d]}$.

We derive a regret lower bound valid for any *uniformly good* algorithm. An algorithm $\pi$ is uniformly good iff $R^\pi(T) = o(T^\alpha)$ for all $\alpha > 0$ and all parameters $\theta \in \Theta$. The proof of this result relies on a general result on controlled Markov chains [19].

**Theorem 1** *For all $\theta \in \Theta$, for any uniformly good policy $\pi \in \Pi$, $\liminf_{T \to \infty} \frac{R^\pi(T)}{\log(T)} \geq c(\theta)$, where $c(\theta)$ is the optimal value of the optimization problem:*

$$\inf_{x \in \mathcal{X}} \sum_{M \in \mathcal{M}} x_M (M^\star(\theta) - M)^\top \theta \qquad s.t. \quad \left(\sum_{M \in \mathcal{M}} x_M M\right)^\top \mathrm{kl}(\theta, \lambda) \geq 1 \, , \, \forall \lambda \in B(\theta). \qquad (1)$$

Observe first that optimization problem (1) is a semi-infinite linear program which can be solved for any fixed $\theta$, but its optimal value is difficult to compute explicitly. Determining how $c(\theta)$ scales as a function of the problem dimensions $d$ and $m$ is not obvious. Also note that (1) has the following interpretation: assume that (1) has a unique solution $x^\star$. Then any uniformly good algorithm must select action $M$ at least $x_M^\star \log(T)$ times over the $T$ first rounds. From [19], we know that there exists an algorithm which is asymptotically optimal, so that its regret matches the lower bound of Theorem 1. However this algorithm suffers from two problems: it is computationally infeasible for large problems since it involves solving (1) $T$ times, furthermore the algorithm has no finite time performance guarantees, and numerical experiments suggest that its finite time performance on typical problems is rather poor. Further remark that if $\mathcal{M}$ is the set of singletons (classical bandit), Theorem 1 reduces to the Lai-Robbins bound [20] and if $\mathcal{M}$ is the set of $m$-sets (bandit with multiple plays), Theorem 1 reduces to the lower bound derived in [6]. Finally, Theorem 1 can be generalized in a straightforward manner for when rewards belong to a one-parameter exponential family of distributions (e.g., Gaussian, Exponential, Gamma etc.) by replacing $\mathrm{kl}$ by the appropriate divergence measure.

**A Simplified Lower Bound**   We now study how the regret $c(\theta)$ scales as a function of the problem dimensions $d$ and $m$. To this aim, we present a simplified regret lower bound. Given $\theta$, we say that a set $\mathcal{H} \subset \mathcal{M} \setminus M^\star$ has property $P(\theta)$ iff, for all $(M, M') \in \mathcal{H}^2$, $M \neq M'$ we have $M_i M_i'(1 - M_i^\star(\theta)) = 0$ for all $i$. We may now state Theorem 2.

**Theorem 2** *Let $\mathcal{H}$ be a maximal (inclusion-wise) subset of $\mathcal{M}$ with property $P(\theta)$. Define $\beta(\theta) = \min_{M \neq M^\star} \frac{\Delta_M}{|M \setminus M^\star|}$. Then:*

$$c(\theta) \geq \sum_{M \in \mathcal{H}} \frac{\beta(\theta)}{\max_{i \in M \setminus M^\star} \mathrm{kl}\left(\theta_i, \frac{1}{|M \setminus M^\star|} \sum_{j \in M^\star \setminus M} \theta_j\right)}.$$

**Corollary 1** *Let $\theta \in [a, 1]^d$ for some constant $a > 0$ and $\mathcal{M}$ be such that each arm $M \in \mathcal{M}, M \neq M^\star$ has at most $k$ suboptimal basic actions. Then $c(\theta) = \Omega(|\mathcal{H}|/k)$.*

Theorem 2 provides an explicit regret lower bound. Corollary 1 states that $c(\theta)$ scales at least with the size of $\mathcal{H}$. For most combinatorial sets, $|\mathcal{H}|$ is proportional to $d - m$ (see supplementary material for some examples), which implies that in these cases, one cannot obtain a regret smaller than $\mathcal{O}((d - m)\Delta_{\min}^{-1} \log(T))$. This result is intuitive since $d - m$ is the number of parameters not observed when selecting the optimal arm. The algorithms proposed below have a regret of $\mathcal{O}(d\sqrt{m}\Delta_{\min}^{-1} \log(T))$, which is acceptable since typically, $\sqrt{m}$ is much smaller than $d$.

## 4.2   Algorithms

Next we present ESCB, an algorithm for stochastic combinatorial bandits that relies on arm indexes as in UCB1 [21] and KL-UCB [5]. We derive finite-time regret upper bounds for ESCB that hold even if we assume that $\|M\|_1 \leq m,\ \forall M \in \mathcal{M}$, instead of $\|M\|_1 = m$, so that arms may have different numbers of basic actions.

### 4.2.1   Indexes

ESCB relies on arm indexes. In general, an index of arm $M$ in round $n$, say $b_M(n)$, should be defined so that $b_M(n) \geq M^\top \theta$ with high probability. Then as for UCB1 and KL-UCB, applying the principle of optimism in face of uncertainty, a natural way to devise algorithms based on indexes is to select in each round the arm with the highest index. Under a given algorithm, at time $n$, we define $t_i(n) = \sum_{s=1}^{n} M_i(s)$ the number of times basic action $i$ has been sampled. The empirical mean reward of action $i$ is then defined as $\hat{\theta}_i(n) = (1/t_i(n)) \sum_{s=1}^{n} X_i(s) M_i(s)$ if $t_i(n) > 0$ and $\hat{\theta}_i(n) = 0$ otherwise. We define the corresponding vectors $t(n) = (t_i(n))_{i \in [d]}$ and $\hat{\theta}(n) = (\hat{\theta}_i(n))_{i \in [d]}$.

The indexes we propose are functions of the round $n$ and of $\hat{\theta}(n)$. Our first index for arm $M$, referred to as $b_M(n, \hat{\theta}(n))$ or $b_M(n)$ for short, is an extension of KL-UCB index. Let $f(n) = \log(n) + 4m \log(\log(n))$. $b_M(n, \hat{\theta}(n))$ is the optimal value of the following optimization problem:

$$\max_{q \in \Theta} M^\top q \qquad \text{s.t.} \quad (Mt(n))^\top \mathrm{kl}(\hat{\theta}(n), q) \leq f(n), \tag{2}$$

where we use the convention that for $v, u \in \mathbb{R}^d$, $vu = (v_i u_i)_{i \in [d]}$. As we show later, $b_M(n)$ may be computed efficiently using a line search procedure similar to that used to determine KL-UCB index.

Our second index $c_M(n, \hat{\theta}(n))$ or $c_M(n)$ for short is a generalization of the UCB1 and UCB-tuned indexes:

$$c_M(n) = M^\top \hat{\theta}(n) + \sqrt{\frac{f(n)}{2}\left(\sum_{i=1}^{d} \frac{M_i}{t_i(n)}\right)}$$

Note that, in the classical bandit problems with independent arms, i.e., when $m = 1$, $b_M(n)$ reduces to the KL-UCB index (which yields an asymptotically optimal algorithm) and $c_M(n)$ reduces to the UCB-tuned index. The next theorem provides generic properties of our indexes. An important consequence of these properties is that the expected number of times where $b_{M^\star}(n, \hat{\theta}(n))$ or $c_{M^\star}(n, \hat{\theta}(n))$ underestimate $\mu^\star(\theta)$ is *finite*, as stated in the corollary below.

**Theorem 3** *(i) For all $n \geq 1$, $M \in \mathcal{M}$ and $\tau \in [0,1]^d$, we have $b_M(n, \tau) \leq c_M(n, \tau)$.*
*(ii) There exists $C_m > 0$ depending on $m$ only such that, for all $M \in \mathcal{M}$ and $n \geq 2$:*

$$\mathbb{P}[b_M(n, \hat{\theta}(n)) \leq M^\top \theta] \leq C_m n^{-1} (\log(n))^{-2}.$$

**Corollary 2** $\sum_{n \geq 1} \mathbb{P}[b_{M^\star}(n, \hat{\theta}(n)) \leq \mu^\star] \leq 1 + C_m \sum_{n \geq 2} n^{-1} (\log(n))^{-2} < \infty.$

Statement (i) in the above theorem is obtained combining Pinsker and Cauchy-Schwarz inequalities. The proof of statement (ii) is based on a concentration inequality on sums of empirical KL divergences proven in [22]. It enables to control the fluctuations of multivariate empirical distributions for exponential families. It should also be observed that indexes $b_M(n)$ and $c_M(n)$ can be extended in a straightforward manner to the case of continuous linear bandit problems, where the set of arms is the unit sphere and one wants to maximize the dot product between the arm and an unknown vector. $b_M(n)$ can also be extended to the case where reward distributions are not Bernoulli but lie in an exponential family (e.g. Gaussian, Exponential, Gamma, etc.), replacing kl by a suitably chosen divergence measure. A close look at $c_M(n)$ reveals that the indexes proposed in [10], [11], and [9] are too conservative to be optimal in our setting: there the "confidence bonus" $\sum_{i=1}^d \frac{M_i}{t_i(n)}$ was replaced by (at least) $m \sum_{i=1}^d \frac{M_i}{t_i(n)}$. Note that [10], [11] assume that the various basic actions are arbitrarily correlated, while we assume independence among basic actions. When independence does not hold, [11] provides a problem instance where the regret is at least $\Omega(\frac{md}{\Delta_{\min}} \log(T))$. This does not contradict our regret upper bound (scaling as $\mathcal{O}(\frac{d\sqrt{m}}{\Delta_{\min}} \log(T))$), since we have added the independence assumption.

### 4.2.2 Index computation

While the index $c_M(n)$ is explicit, $b_M(n)$ is defined as the solution to an optimization problem. We show that it may be computed by a simple line search. For $\lambda \geq 0$, $w \in [0,1]$ and $v \in \mathbb{N}$, define:

$$g(\lambda, w, v) = \left(1 - \lambda v + \sqrt{(1 - \lambda v)^2 + 4wv\lambda}\right) / 2.$$

Fix $n$, $M$, $\hat{\theta}(n)$ and $t(n)$. Define $I = \{i : M_i = 1, \hat{\theta}_i(n) \neq 1\}$, and for $\lambda > 0$, define:

$$F(\lambda) = \sum_{i \in I} t_i(n) \mathrm{kl}(\hat{\theta}_i(n), g(\lambda, \hat{\theta}_i(n), t_i(n))).$$

**Theorem 4** *If $I = \emptyset$, $b_M(n) = ||M||_1$. Otherwise: (i) $\lambda \mapsto F(\lambda)$ is strictly increasing, and $F(\mathbb{R}^+) = \mathbb{R}^+$. (ii) Define $\lambda^\star$ as the unique solution to $F(\lambda) = f(n)$. Then $b_M(n) = ||M||_1 - |I| + \sum_{i \in I} g(\lambda^\star, \hat{\theta}_i(n), t_i(n))$.*

Theorem 4 shows that $b_M(n)$ can be computed using a line search procedure such as bisection, as this computation amounts to solving the nonlinear equation $F(\lambda) = f(n)$, where $F$ is strictly increasing. The proof of Theorem 4 follows from KKT conditions and the convexity of the KL divergence.

### 4.2.3 The ESCB Algorithm

The pseudo-code of ESCB is presented in Algorithm 1. We consider two variants of the algorithm based on the choice of the index $\xi_M(n)$: ESCB-1 when $\xi_M(n) = b_M(n)$ and ESCB-2 if $\xi_M(n) = c_M(n)$. In practice, ESCB-1 outperforms ESCB-2. Introducing ESCB-2 is however instrumental in the regret analysis of ESCB-1 (in view of Theorem 3 (i)). The following theorem provides a finite time analysis of our ESCB algorithms. The proof of this theorem borrows some ideas from the proof of [11, Theorem 3].

**Theorem 5** *The regret under algorithms $\pi \in \{\text{ESCB-1}, \text{ESCB-2}\}$ satisfies for all $T \geq 1$:*

$$R^\pi(T) \leq 16d\sqrt{m}\Delta_{\min}^{-1} f(T) + 4dm^3\Delta_{\min}^{-2} + C_m',$$

*where $C_m' \geq 0$ does not depend on $\theta$, $d$ and $T$. As a consequence $R^\pi(T) = \mathcal{O}(d\sqrt{m}\Delta_{\min}^{-1} \log(T))$ when $T \to \infty$.*

---

**Algorithm 1** ESCB

> **for** $n \geq 1$ **do**
>> Select arm $M(n) \in \arg \max_{M \in \mathcal{M}} \xi_M(n)$.
>> Observe the rewards, and update $t_i(n)$ and $\hat{\theta}_i(n), \forall i \in M(n)$.
> **end for**

---

**Algorithm 2** COMBEXP

> **Initialization:** Set $q_0 = \mu^0, \gamma = \frac{\sqrt{m \log \mu_{\min}^{-1}}}{\sqrt{m \log \mu_{\min}^{-1}} + \sqrt{C(Cm^2 d + m)T}}$ and $\eta = \gamma C$, with $C = \frac{\lambda}{m^{3/2}}$.
> **for** $n \geq 1$ **do**
>> *Mixing:* Let $q'_{n-1} = (1 - \gamma) q_{n-1} + \gamma \mu^0$.
>> *Decomposition:* Select a distribution $p_{n-1}$ over $\mathcal{M}$ such that $\sum_M p_{n-1}(M) M = m q'_{n-1}$.
>> *Sampling:* Select a random arm $M(n)$ with distribution $p_{n-1}$ and incur a reward $Y_n = \sum_i X_i(n) M_i(n)$.
>> *Estimation:* Let $\Sigma_{n-1} = \mathbb{E}\left[MM^\top\right]$, where $M$ has law $p_{n-1}$. Set $\tilde{X}(n) = Y_n \Sigma_{n-1}^+ M(n)$, where $\Sigma_{n-1}^+$ is the pseudo-inverse of $\Sigma_{n-1}$.
>> *Update:* Set $\tilde{q}_n(i) \propto q_{n-1}(i) \exp(\eta \tilde{X}_i(n)), \; \forall i \in [d]$.
>> *Projection:* Set $q_n$ to be the projection of $\tilde{q}_n$ onto the set $\mathcal{P}$ using the KL divergence.
> **end for**

---

ESCB with time horizon $T$ has a complexity of $\mathcal{O}(|\mathcal{M}|T)$ as neither $b_M$ nor $c_M$ can be written as $M^\top y$ for some vector $y \in \mathbb{R}^d$. Assuming that the offline (static) combinatorial problem is solvable in $\mathcal{O}(V(\mathcal{M}))$ time, the complexity of the CUCB algorithm in [10] and [11] after $T$ rounds is $\mathcal{O}(V(\mathcal{M})T)$. Thus, if the offline problem is efficiently implementable, i.e., $V(\mathcal{M}) = \mathcal{O}(\text{poly}(d/m))$, CUCB is efficient, whereas ESCB is not since $|\mathcal{M}|$ may have exponentially many elements. In §2.5 of the supplement, we provide an extension of ESCB called EPOCH-ESCB, that attains almost the same regret as ESCB while enjoying much better computational complexity.

## 5 Adversarial Combinatorial Bandits under Bandit Feedback

We now consider adversarial combinatorial bandits with bandit feedback. We start with the following observation:

$$\max_{M \in \mathcal{M}} M^\top X = \max_{\mu \in Co(\mathcal{M})} \mu^\top X,$$

with $Co(\mathcal{M})$ the convex hull of $\mathcal{M}$. We embed $\mathcal{M}$ in the $d$-dimensional simplex by dividing its elements by $m$. Let $\mathcal{P}$ be this scaled version of $Co(\mathcal{M})$.

Inspired by OSMD [13, 18], we propose the COMBEXP algorithm, where the KL divergence is the Bregman divergence used to project onto $\mathcal{P}$. Projection using the KL divergence is addressed in [23]. We denote the KL divergence between distributions $q$ and $p$ in $\mathcal{P}$ by $\text{KL}(p, q) = \sum_{i \in [d]} p(i) \log \frac{p(i)}{q(i)}$. The projection of distribution $q$ onto a closed convex set $\Xi$ of distributions is $p^\star = \arg \min_{p \in \Xi} \text{KL}(p, q)$.

Let $\underline{\lambda}$ be the smallest nonzero eigenvalue of $\mathbb{E}[MM^\top]$, where $M$ is uniformly distributed over $\mathcal{M}$. We define the exploration-inducing distribution $\mu^0 \in \mathcal{P}$: $\mu_i^0 = \frac{1}{m|\mathcal{M}|} \sum_{M \in \mathcal{M}} M_i, \quad \forall i \in [d]$, and let $\mu_{\min} = \min_i m \mu_i^0$. $\mu^0$ is the distribution over basic actions $[d]$ induced by the uniform distribution over $\mathcal{M}$. The pseudo-code for COMBEXP is shown in Algorithm 2. The KL projection in COMBEXP ensures that $m q_{n-1} \in Co(\mathcal{M})$. There exists $\lambda$, a distribution over $\mathcal{M}$ such that $m q_{n-1} = \sum_M \lambda(M) M$. This guarantees that the system of linear equations in the *decomposition step* is consistent. We propose to perform the *projection step* (the KL projection of $\tilde{q}$ onto $\mathcal{P}$) using interior-point methods [24]. We provide a simpler method in §3.4 of the supplement. The *decomposition step* can be efficiently implemented using the algorithm of [25]. The following theorem provides a regret upper bound for COMBEXP.

**Theorem 6** *For all $T \geq 1$:* $R^{\text{COMBEXP}}(T) \leq 2\sqrt{m^3 T \left(d + \frac{m^{1/2}}{\underline{\lambda}}\right) \log \mu_{\min}^{-1}} + \frac{m^{5/2}}{\underline{\lambda}} \log \mu_{\min}^{-1}$.

For most classes of $\mathcal{M}$, we have $\mu_{\min}^{-1} = \mathcal{O}(\text{poly}(d/m))$ and $m(d\underline{\lambda})^{-1} = \mathcal{O}(1)$ [4]. For these classes, COMBEXP has a regret of $\mathcal{O}(\sqrt{m^3 dT \log(d/m)})$, which is a factor $\sqrt{m \log(d/m)}$ off the lower bound (see Table 2).

It might not be possible to compute the projection step exactly, and this step can be solved up to accuracy $\epsilon_n$ in round $n$. Namely we find $q_n$ such that $\text{KL}(q_n, \tilde{q}_n) - \min_{p \in \Xi} \text{KL}(p, \tilde{q}_n) \leq \epsilon_n$. Proposition 1 shows that for $\epsilon_n = \mathcal{O}(n^{-2} \log^{-3}(n))$, the approximate projection gives the same regret as when the projection is computed exactly. Theorem 7 gives the computational complexity of COMBEXP with approximate projection. When $Co(\mathcal{M})$ is described by polynomially (in $d$) many linear equalities/inequalities, COMBEXP is efficiently implementable and its running time scales (almost) linearly in $T$. Proposition 1 and Theorem 7 easily extend to other OSMD-type algorithms and thus might be of independent interest.

**Proposition 1** *If the projection step of* COMBEXP *is solved up to accuracy* $\epsilon_n = \mathcal{O}(n^{-2} \log^{-3}(n))$, *we have:*

$$R^{\text{COMBEXP}}(T) \leq 2\sqrt{2m^3 T \left(d + \frac{m^{1/2}}{\underline{\lambda}}\right) \log \mu_{\min}^{-1}} + \frac{2m^{5/2}}{\underline{\lambda}} \log \mu_{\min}^{-1}.$$

**Theorem 7** *Assume that* $Co(\mathcal{M})$ *is defined by* $c$ *linear equalities and* $s$ *linear inequalities. If the projection step is solved up to accuracy* $\epsilon_n = \mathcal{O}(n^{-2} \log^{-3}(n))$*, then* COMBEXP *has time complexity.*

The time complexity of COMBEXP can be reduced by exploiting the structure of $\mathcal{M}$ (See [24, page 545]). In particular, if inequality constraints describing $Co(\mathcal{M})$ are box constraints, the time complexity of COMBEXP is $\mathcal{O}(T[c^2\sqrt{s}(c+d)\log(T) + d^4])$.

The computational complexity of COMBEXP is determined by the structure of $Co(\mathcal{M})$ and COMB-EXP has $\mathcal{O}(T \log(T))$ time complexity due to the efficiency of interior-point methods. In contrast, the computational complexity of COMBAND depends on the complexity of sampling from $\mathcal{M}$. COMBAND may have a time complexity that is super-linear in $T$ (see [16, page 217]). For instance, consider the matching problem described in Section 2. We have $c = 2m$ equality constraints and $s = m^2$ box constraints, so that the time complexity of COMBEXP is: $\mathcal{O}(m^5 T \log(T))$. It is noted that using [26, Algorithm 1], the cost of decomposition in this case is $\mathcal{O}(m^4)$. On the other hand, COMBBAND has a time complexity of $\mathcal{O}(m^{10} F(T))$, with $F$ a super-linear function, as it requires to approximate a permanent, requiring $\mathcal{O}(m^{10})$ operations per round. Thus, COMBEXP has much lower complexity than COMBAND and achieves the same regret.

## 6  Conclusion

We have investigated stochastic and adversarial combinatorial bandits. For stochastic combinatorial bandits with semi-bandit feedback, we have provided a tight, problem-dependent regret lower bound that, in most cases, scales at least as $\mathcal{O}((d - m)\Delta_{\min}^{-1} \log(T))$. We proposed ESCB, an algorithm with $\mathcal{O}(d\sqrt{m}\Delta_{\min}^{-1} \log(T))$ regret. We plan to reduce the gap between this regret guarantee and the regret lower bound, as well as investigate the performance of EPOCH-ESCB. For adversarial combinatorial bandits with bandit feedback, we proposed the COMBEXP algorithm. There is a gap between the regret of COMBEXP and the known regret lower bound in this setting, and we plan to reduce it as much as possible.

**Acknowledgments**

A. Proutiere's research is supported by the ERC FSA grant, and the SSF ICT-Psi project.

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
