[Supplementary Material]

# Combinatorial Bandits Revisited
# Supplementary Material

**Richard Combes**[*]  **M. Sadegh Talebi**[†]  **Alexandre Proutiere**[†]  **Marc Lelarge**[‡]
[*] Centrale-Supelec, L2S, Gif-sur-Yvette, FRANCE
[†] Department of Automatic Control, KTH, Stockholm, SWEDEN
[‡] INRIA & ENS, Paris, FRANCE
richard.combes@supelec.fr,{mstms,alepro}@kth.se,marc.lelarge@ens.fr

## 1 Stochastic Combinatorial Bandits: Regret Lower Bounds

### 1.1 Proof of Theorem 1

To derive regret lower bounds, we apply the techniques used by Graves and Lai [1] to investigate efficient adaptive decision rules in controlled Markov chains. First we give an overview of their general framework.

Consider a controlled Markov chain $(X_n)_{n \geq 0}$ on a finite state space $\mathcal{S}$ with a control set $U$. The transition probabilities given control $u \in U$ are parameterized by $\theta$ taking values in a compact metric space $\Theta$: the probability to move from state $x$ to state $y$ given the control $u$ and the parameter $\theta$ is $p(x, y; u, \theta)$. The parameter $\theta$ is not known. The decision maker is provided with a finite set of stationary control laws $G = \{g_1, \ldots, g_K\}$, where each control law $g_j$ is a mapping from $\mathcal{S}$ to $U$: when control law $g_j$ is applied in state $x$, the applied control is $u = g_j(x)$. It is assumed that if the decision maker always selects the same control law $g$, the Markov chain is then irreducible with stationary distribution $\pi_\theta^g$. Now the reward obtained when applying control $u$ in state $x$ is denoted by $r(x, u)$, so that the expected reward achieved under control law $g$ is: $\mu_\theta(g) = \sum_x r(x, g(x))\pi_\theta^g(x)$. There is an optimal control law given $\theta$ whose expected reward is denoted by $\mu_\theta^\star = \max_{g \in G} \mu_\theta(g)$. Now the objective of the decision maker is to sequentially select control laws so as to maximize the expected reward up to a given time horizon $T$. As for MAB problems, the performance of a decision scheme can be quantified through the notion of regret which compares the expected reward to that obtained by always applying the optimal control law.

**Proof.** The parameter $\theta$ takes values in $[0, 1]^d$. The Markov chain has values in $\mathcal{S} = \{0, 1\}^d$. The set of controls corresponds to the set of feasible actions $\mathcal{M}$, and the set of control laws is also $\mathcal{M}$. These laws are constant, in the sense that the control applied by control law $M \in \mathcal{M}$ does not depend on the state of the Markov chain, and corresponds to selecting action $M$. The transition probabilities are given as follows: for all $x, y \in \mathcal{S}$,

$$p(x, y; M, \theta) = p(y; M, \theta) = \prod_{i \in [d]} p_i(y_i; M, \theta),$$

where for all $i \in [d]$, if $M_i = 0$, $p_i(0; M, \theta) = 1$, and if $M_i = 1$, $p_i(y_i; M, \theta) = \theta_i^{y_i}(1 - \theta_i)^{1 - y_i}$. Finally, the reward $r(y, M)$ is defined by $r(y, M) = M^\top y$. Note that the state space of the Markov chain is here finite, and so, we do not need to impose any cost associated with switching control laws (see the discussion on page 718 in [1]).

We can now apply Theorem 1 in [1]. Note that the KL number under action $M$ is

$$\mathrm{kl}^M(\theta, \lambda) = \sum_{i \in [d]} M_i \mathrm{kl}(\theta_i, \lambda_i).$$

From [1, Theorem 1], we conclude that for any uniformly good rule $\pi$,

$$\liminf_{T \to \infty} \frac{R^\pi(T)}{\log(T)} \geq c(\theta),$$

where $c(\theta)$ is the optimal value of the following optimization problem:

$$\inf_{x_M \geq 0, M \in \mathcal{M}} \sum_{M \neq M^\star} x_M(\mu^\star - \mu_M(\theta)), \tag{1}$$

$$\text{s.t.} \inf_{\lambda \in B(\theta)} \sum_{Q \neq M^\star} x_Q \text{kl}^Q(\theta, \lambda) \geq 1. \tag{2}$$

The result is obtained by observing that $B(\theta) = \bigcup_{M \neq M^\star} B_M(\theta)$, where

$$B_M(\theta) = \{\lambda \in \Theta : M_i^\star(\theta)(\theta_i - \lambda_i) = 0, \forall i, \ \mu^\star(\theta) < \mu_M(\lambda)\}.$$

$\square$

## 1.2 Proof of Theorem 2

The proof proceeds in three steps. In the subsequent analysis, given the optimization problem $\mathsf{P}$, we use $\text{val}(\mathsf{P})$ to denote its optimal value.

**Step 1.** In this step, first we introduce an equivalent formulation for problem (1) above by simplifying its constraints. We show that constraint (2) is equivalent to:

$$\inf_{\lambda \in B_M(\theta)} \sum_{i \in M \setminus M^\star} \text{kl}(\theta_i, \lambda_i) \sum_{Q \in \mathcal{M}} Q_i x_Q \geq 1, \ \forall M \neq M^\star.$$

Observe that:

$$\sum_{Q \neq M^\star} x_Q \text{kl}^Q(\theta, \lambda) = \sum_{Q \neq M^\star} x_Q \sum_{i \in [d]} Q_i \text{kl}(\theta_i, \lambda_i) = \sum_{i \in [d]} \text{kl}(\theta_i, \lambda_i) \sum_{Q \neq M^\star} Q_i x_Q.$$

Fix $M \neq M^\star$. In view of the definition of $B_M(\theta)$, we can find $\lambda \in B_M(\theta)$ such that $\lambda_i = \theta_i, \forall i \in ([d] \setminus M) \cup M^\star$. Thus, for the r.h.s. of the $M$-th constraint in (2), we get:

$$\inf_{\lambda \in B_M(\theta)} \sum_{Q \neq M^\star} x_Q \text{kl}^Q(\theta, \lambda) = \inf_{\lambda \in B_M(\theta)} \sum_{i \in [d]} \text{kl}(\theta_i, \lambda_i) \sum_{Q \neq M^\star} Q_i x_Q$$

$$= \inf_{\lambda \in B_M(\theta)} \sum_{i \in M \setminus M^\star} \text{kl}(\theta_i, \lambda_i) \sum_Q Q_i x_Q,$$

and therefore problem (1) can be equivalently written as:

$$c(\theta) = \inf_{x_M \geq 0, M \in \mathcal{M}} \sum_{M \neq M^\star} x_M(\mu^\star - \mu_M(\theta)), \tag{3}$$

$$\text{s.t.} \inf_{\lambda \in B_M(\theta)} \sum_{i \in M \setminus M^\star} \text{kl}(\theta_i, \lambda_i) \sum_Q Q_i x_Q \geq 1, \ \forall M \neq M^\star. \tag{4}$$

Next, we formulate an LP whose value gives a lower bound for $c(\theta)$. Define $\hat{\lambda}(M) = (\hat{\lambda}_i(M), i \in [d])$ with

$$\hat{\lambda}_i(M) = \begin{cases} \frac{1}{|M \setminus M^\star|} \sum_{j \in M^\star \setminus M} \theta_j & \text{if } i \in M \setminus M^\star, \\ \theta_i & \text{otherwise.} \end{cases}$$

Clearly $\hat{\lambda}(M) \in B_M(\theta)$, and therefore:

$$\inf_{\lambda \in B_M(\theta)} \sum_{i \in M \setminus M^\star} \text{kl}(\theta_i, \lambda_i) \sum_Q Q_i x_Q \leq \sum_{i \in M \setminus M^\star} \text{kl}(\theta_i, \hat{\lambda}_i(M)) \sum_Q Q_i x_Q,$$

Then, we can write:

$$c(\theta) \geq \inf_{x \geq 0} \sum_{M \neq M^\star} \Delta_M x_M \tag{5}$$

$$\text{s.t.} \quad \sum_{i \in M \setminus M^\star} \text{kl}(\theta_i, \hat{\lambda}_i(M)) \sum_Q Q_i x_Q \geq 1, \quad \forall M \neq M^\star. \tag{6}$$

For any $M \neq M^\star$ introduce: $g_M = \max_{i \in M \setminus M^\star} \text{kl}(\theta_i, \hat{\lambda}_i(M))$. Now we form P1 as follows:

$$\text{P1:} \quad \inf_{x \geq 0} \sum_{M \neq M^\star} \Delta_M x_M \tag{7}$$

$$\text{s.t.} \quad \sum_{i \in M \setminus M^\star} \sum_Q Q_i x_Q \geq \frac{1}{g_M}, \quad \forall M \neq M^\star. \tag{8}$$

Observe that $c(\theta) \geq \text{val}(\text{P1})$ since the feasible set of problem (5) is contained in that of P1.

**Step 2.** In this step, we formulate an LP to give a lower bound for $\text{val}(\text{P1})$. To this end, for any suboptimal basic action $i \in [d]$, we define $z_i = \sum_M M_i x_M$. Further, we let $z = [z_i, i \in [d]]$. Next, we represent the objective of P1 in terms of $z$, and give a lower bound for it as follows:

$$
\begin{aligned}
\sum_{M \neq M^\star} \Delta_M x_M &= \sum_{M \neq M^\star} x_M \sum_{i \in M \setminus M^\star} \frac{\Delta_M}{|M \setminus M^\star|} \\
&= \sum_{M \neq M^\star} x_M \sum_{i \in [d] \setminus M^\star} \frac{\Delta_M}{|M \setminus M^\star|} M_i \\
&\geq \min_{M \neq M^\star} \frac{\Delta_M}{|M \setminus M^\star|} \cdot \sum_{i \in [d] \setminus M^\star} \sum_{M' \neq M^\star} M'_i x_{M'} \\
&= \min_{M \neq M^\star} \frac{\Delta_M}{|M \setminus M^\star|} \cdot \sum_{i \in [d] \setminus M^\star} z_i \\
&= \beta(\theta) \sum_{i \in [d] \setminus M^\star} z_i.
\end{aligned}
$$

Then, defining

$$\text{P2:} \quad \inf_{z \geq 0} \beta(\theta) \sum_{i \in [d] \setminus M^\star} z_i$$

$$\text{s.t.} \quad \sum_{i \in M \setminus M^\star} z_i \geq \frac{1}{g_M}, \quad \forall M \neq M^\star,$$

yields: $\text{val}(\text{P1}) \geq \text{val}(\text{P2})$.

**Step 3.** Introduce set $\mathcal{H}$ satisfying property $P(\theta)$ as stated in Section 4. Now define

$$\mathcal{Z} = \left\{ z \in \mathbb{R}^d_+ : \sum_{i \in M \setminus M^\star} z_i \geq \frac{1}{g_M}, \quad \forall M \in \mathcal{H} \right\},$$

and

$$\text{P3:} \quad \inf_{z \in \mathcal{Z}} \beta(\theta) \sum_{i \in [d] \setminus M^\star} z_i.$$

Observe that $\mathrm{val}(\mathsf{P2}) \geq \mathrm{val}(\mathsf{P3})$ since the feasible set of $\mathsf{P2}$ is contained in $\mathcal{Z}$. The definition of $\mathcal{H}$ implies that $\sum_{i \in [d] \setminus M^\star} z_i = \sum_{M \in \mathcal{H}} \sum_{i \in M \setminus M^\star} z_i$. It then follows that

$$
\begin{aligned}
\mathrm{val}(\mathsf{P3}) &= \sum_{M \in \mathcal{H}} \frac{\beta(\theta)}{g_M} \\
&\geq \sum_{M \in \mathcal{H}} \frac{\beta(\theta)}{\max_{i \in M \setminus M^\star} \mathrm{kl}(\theta_i, \hat{\lambda}_i(M))} \\
&= \sum_{M \in \mathcal{H}} \frac{\beta(\theta)}{\max_{i \in M \setminus M^\star} \mathrm{kl}\left(\theta_i, \frac{1}{|M \setminus M^\star|} \sum_{j \in M^\star \setminus M} \theta_j\right)}.
\end{aligned}
$$

The proof is completed by observing that: $c(\theta) \geq \mathrm{val}(\mathsf{P1}) \geq \mathrm{val}(\mathsf{P2}) \geq \mathrm{val}(\mathsf{P3})$. $\qquad\square$

### 1.3 Proof of Corollary 1

Fix $M \neq M^\star$. For any $i \in M \setminus M^\star$, we have:

$$
\begin{aligned}
\mathrm{kl}\left(\theta_i, \frac{1}{|M \setminus M^\star|} \sum_{j \in M^\star \setminus M} \theta_j\right) &\leq \frac{1}{|M \setminus M^\star|} \sum_{j \in M^\star \setminus M} \mathrm{kl}\left(\theta_i, \theta_j\right) \quad \text{(By convexity of } \mathrm{kl}(.,.)\text{)} \\
&\leq \frac{1}{|M \setminus M^\star|} \sum_{j \in M^\star \setminus M} \frac{(\theta_i - \theta_j)^2}{\theta_j (1 - \theta_j)} \\
&\leq \frac{1}{|M \setminus M^\star|} \sum_{j \in M^\star \setminus M} \frac{(1 - \theta_j)^2}{\theta_j (1 - \theta_j)} \\
&\leq \frac{1}{|M \setminus M^\star|} \sum_{j \in M^\star \setminus M} \left(\frac{1}{\theta_j} - 1\right) \\
&\leq \frac{1}{\min_{j \in M^\star \setminus M} \theta_j} - 1 \\
&\leq \frac{1}{a} - 1,
\end{aligned}
$$

where the second inequality follows from the inequality $\mathrm{kl}(p, q) \leq \frac{(p-q)^2}{q(1-q)}$ for all $(p, q) \in [0, 1]^2$. Moreover, we have that

$$
\beta(\theta) = \min_{M \neq M^\star} \frac{\Delta^M}{|M \setminus M^\star|} \geq \frac{\Delta_{\min}}{\max_M |M \setminus M^\star|} = \frac{\Delta_{\min}}{k}.
$$

Applying Theorem 2, we get:

$$
c(\theta) \geq \sum_{M \in \mathcal{H}} \frac{\beta(\theta)}{\max_{i \in M \setminus M^\star} \mathrm{kl}\left(\theta_i, \frac{1}{|M \setminus M^\star|} \sum_{j \in M^\star \setminus M} \theta_j\right)} \geq \frac{\Delta_{\min} a}{k(1 - a)} |\mathcal{H}|,
$$

which gives the required lower bound and completes the proof. $\qquad\square$

### 1.4 Examples of Scaling of the Lower Bound

#### 1.4.1 Matchings

In the first example, we assume that $\mathcal{M}$ is the set of perfect matchings in the complete bipartite graph $\mathcal{K}_{m,m}$, with $|\mathcal{M}| = m!$ and $d = m^2$. A maximal subset $\mathcal{H}$ of $\mathcal{M}$ satisfying property $P(\theta)$ can be constructed by adding all matchings that differ from the optimal matching by only two edges, see Figure 1 for illustration in the case of $m = 4$. Here $|\mathcal{H}| = \binom{m}{2}$ and thus, $|\mathcal{H}|$ scales as $m^2 = d$.

Figure 1: Matchings in $\mathcal{K}_{4,4}$: (a) The optimal matching $M^\star$, (b)-(g) Elements of $\mathcal{H}$.

Figure 2: Spanning trees in $\mathcal{K}_5$: (a) The optimal spanning tree $M^\star$, (b)-(g) Elements of $\mathcal{H}$.

### 1.4.2 Spanning trees

Consider the problem of finding the minimum spanning tree in a complete graph $\mathcal{K}_N$. This corresponds to letting $\mathcal{M}$ be the set of all spanning trees in $\mathcal{K}_N$, where $|\mathcal{M}| = N^{N-2}$ (Cayley's formula). In this case, we have $d = \binom{N}{2} = \frac{N(N-1)}{2}$, which is the number of edges of $\mathcal{K}_N$, and $m = N - 1$. A maximal subset $\mathcal{H}$ of $\mathcal{M}$ satisfying property $P(\theta)$ can be constructed by composing all spanning trees that differ from the optimal tree by one edge only, see Figure 2. In this case, $\mathcal{H}$ has $d - m = \frac{(N-1)(N-2)}{2}$ elements.

### 1.4.3 Routing in a grid

Now we give an example, in which $|\mathcal{H}|$ is not scaling as $\Omega(d)$. Consider routing in an $N$-by-$N$ directed grid, whose topology is shown in Figure 3(a) where the source (resp. destination) node is shown in red (resp. blue). Here $\mathcal{M}$ is the set of all $\binom{2N-2}{N-1}$ paths with $m = 2(N-1)$ edges. We further have $d = 2N(N-1)$. In this example, elements of any maximal set $\mathcal{H}$ satisfying $P(\theta)$ do not cover all basic actions. For instance, for the grid shown in Figure 3(a), the two edges incident to the right lower corner do not appear in any arm in $\mathcal{H}$. It can be easily verified that in this case, $|\mathcal{H}|$ scales as $N$ rather than $N^2 = d$.

### 1.5 Lower Bound Example

Here we provide an example, motivated by [2], to investigate the tightness of the regret bounds of our algorithms. Consider the topology shown in Figure 4, where there are $\frac{d}{m}$ paths, each consisting of $m$ links. Let parameter $\theta$ be defined such that

$$\theta_i = \begin{cases} 0.5 & \text{if } i \text{ belongs to the first path} \\ 0.5 - \delta & \text{otherwise.} \end{cases}$$

The first path is the optimal path and for any $M \neq 1$ we have: $\Delta_M = \Delta = m\delta$. Since various paths are independent, this problem reduces to a classical MAB problem with $\frac{d}{m}$ arms. It is observed that the total reward of each path is the sum of $m$ independent Bernoulli random variables with the same parameter. Hence, it is distributed according to a binomial distribution. It then follows that

Figure 3: Routing in a grid: (a) Grid topology with source (red) and destination (blue) nodes, (b) Optimal path $M^\star$, (c)-(e) Elements of $\mathcal{H}$.

Figure 4: Lower bound example

$$\liminf_{T \to \infty} \frac{R(T)}{\log(T)} \geq \sum_{M \neq M^\star} \frac{\Delta_M}{\text{KL}(\text{Bin}(m, 0.5 - \delta), \text{Bin}(m, 0.5))}$$

$$= \left( \frac{d}{m} - 1 \right) \cdot \frac{\Delta}{m \text{kl}(0.5 - \delta, 0.5)}$$

$$\geq \frac{(d - m)\Delta}{4m^2 \delta^2}$$

$$= \frac{d - m}{4\Delta},$$

where the first equality follows from the fact that the KL divergence between two Binomial distributions with respective parameters $(m, u)$ and $(m, v)$ is $m \text{kl}(u, v)$, and where the last step is due to inequality $\text{kl}(x, y) \leq \frac{(x - y)^2}{y(1 - y)}$ for all $x, y \in (0, 1)$.

## 2 Stochastic Combinatorial Bandits: Regret Analysis of ESCB

We use the convention that for $v, u \in \mathbb{R}^d$, $vu = (v_i u_i)_{i \in [d]}$.

### 2.1 A concentration inequality

We first recall Lemma 1, a concentration inequality derived in [3, Theorem 2].

**Lemma 1** *There exists a number $C_m > 0$ depending only on $m$ such that, for all $M$ and all $n \geq 2$:*

$$\mathbb{P}[(Mt(n))^\top \mathrm{kl}(\hat{\theta}(n), \theta) \geq f(n)] \leq C_m n^{-1} (\log(n))^{-2}.$$

### 2.2 Proof of Theorem 3

**First statement:**

Consider $q \in \Theta$, and apply the Cauchy-Schwartz inequality:

$$M^\top(q - \hat{\theta}(n)) = \sum_{i=1}^d \sqrt{t_i(n)}(q_i - \hat{\theta}_i(n)) \frac{M_i}{\sqrt{t_i(n)}} \leq \sqrt{\sum_{i=1}^d M_i t_i(n)(q_i - \hat{\theta}_i(n))^2} \sqrt{\sum_{i=1}^d \frac{M_i}{t_i(n)}}$$

By Pinsker's inequality, for all $(p, q) \in [0, 1]^2$ we have $2(p - q)^2 \leq \mathrm{kl}(p, q)$ so that:

$$M^\top(q - \hat{\theta}(n)) \leq \sqrt{\frac{(Mt(n))^\top \mathrm{kl}(\hat{\theta}(n), q)}{2}} \sqrt{\sum_{i=1}^d \frac{M_i}{t_i(n)}}$$

Hence, $(Mt(n))^\top \mathrm{kl}(\hat{\theta}(n), q) \leq f(n)$ implies:

$$M^\top q = M^\top \hat{\theta}(n) + M^\top(q - \hat{\theta}(n)) \leq M^\top \hat{\theta}(n) + \sqrt{\frac{f(n)}{2} \sum_{i=1}^d \frac{M_i}{t_i(n)}} = c_M(n).$$

so that, by definition of $b_M(n)$, we have $b_M(n) \leq c_M(n)$.

**Second statement:**

If $(Mt(n))^\top \mathrm{kl}(\hat{\theta}(n), \theta) \leq f(n)$ then, by definition of $b_M(n)$ we have $b_M(n) \geq M^\top \theta$. Therefore, using Lemma 1, there exists $C_m$ such that for all $n \geq 2$ we have:

$$\mathbb{P}[b_M(n) < M^\top \theta] \leq \mathbb{P}[(Mt(n))^\top \mathrm{kl}(\hat{\theta}(n), \theta) \geq f(n)] \leq C_m n^{-1} (\log(n))^{-2},$$

which concludes the proof.

### 2.3 Proof of Theorem 4

We recall the following facts about the KL divergence kl, for all $p \in [0, 1]$:

(i) $q \mapsto \mathrm{kl}(p, q)$ is strictly convex on $[0, 1]$ and attains its minimum at $p$, with $\mathrm{kl}(p, p) = 0$.

(ii) Its derivative with respect to the second parameter $q \mapsto \mathrm{kl}'(p, q) = \frac{q-p}{q(1-q)}$ is strictly increasing on $(p, 1)$.

(iii) For $p < 1$, we have $\mathrm{kl}(p, q) \underset{q \to 1^-}{\to} \infty$ and $\mathrm{kl}'(p, q) \underset{q \to 1^-}{\to} \infty$.

Consider $M$ and $n$ fixed throughout the proof. Define $I = \{i \in M : \hat{\theta}_i(n) \neq 1\}$. Consider $q^\star \in \Theta$ the optimal solution of optimization problem:

$$\max_{q \in \Theta} M^\top q$$

$$\text{s.t. } (Mt(n))^\top \mathrm{kl}(\hat{\theta}(n), q) \leq f(n).$$

so that $b_M(n) = M^\top q^\star$. Consider $i \notin M$, then $M^\top q$ does not depend on $q_i$ and from (i) we get $q_i = \hat{\theta}_i(n)$. Now consider $i \in M$. From (i) we get that $1 \geq q_i^\star \geq \hat{\theta}_i(n)$. Hence $q_i^\star = 1$ if $\hat{\theta}_i(n) = 1$. If $I$ is empty, then $q_i^\star = 1$ for all $i \in M$, so that $b_M(n) = ||M||_1$.

Consider the case where $I \neq \emptyset$. From (iii) and the fact that $t(n)^\top \mathrm{kl}(\hat{\theta}(n), q^\star) < \infty$ we get $\hat{\theta}_i(n) \leq q_i^\star < 1$. From the Karush-Kuhn-Tucker (KKT) conditions, there exists $\lambda^\star > 0$ such that for all $i \in I$:

$$1 = \lambda^\star t_i(n)\mathrm{kl}'(\hat{\theta}_i(n), q_i^\star).$$

For $\lambda > 0$ define $\hat{\theta}_i(n) \leq \overline{q}_i(\lambda) < 1$ a solution to the equation:

$$1 = \lambda t_i(n)\mathrm{kl}'(\hat{\theta}_i(n), \overline{q}_i(\lambda)).$$

From (i) we have that $\lambda \mapsto \overline{q}_i(\lambda)$ is uniquely defined, is strictly decreasing and $\hat{\theta}_i(n) < \overline{q}_i(\lambda) < 1$. From (iii) we get that $\overline{q}_i(\mathbb{R}^+) = [\hat{\theta}_i(n), 1]$. Define the function:

$$F(\lambda) = \sum_{i \in I} t_i(n)\mathrm{kl}(\hat{\theta}(n), \overline{q}_i(\lambda)).$$

From the reasoning below, $F$ is well defined, strictly increasing and $F(\mathbb{R}^+) = \mathbb{R}^+$. Therefore, $\lambda^\star$ is the unique solution to $F(\lambda^\star) = f(n)$, and $q_i^\star = \overline{q}_i(\lambda^\star)$. Furthermore, replacing $\mathrm{kl}'$ by its expression we obtain the quadratic equation:

$$\overline{q}_i(\lambda)^2 + \overline{q}_i(\lambda)(\lambda t_i(n) - 1) - \lambda t_i(n)\hat{\theta}_i(n) = 0.$$

Solving for $\overline{q}_i(\lambda)$, we obtain that $\overline{q}_i(\lambda) = g(\lambda, \hat{\theta}_i(n), t_i(n))$, which concludes the proof. $\qquad\square$

## 2.4 Proof of Theorem 5

To prove Theorem 5, we borrow some ideas from proof of [2, Theorem 3].

For any $n \in \mathbb{N}$, $s \in \mathbb{R}^d$, and $M \in \mathcal{M}$ define $h_{n,s,M} = \sqrt{\frac{f(n)}{2}\sum_{i=1}^d \frac{M_i}{s_i}}$, and introduce the following events:

$$G_n = \{(M^\star t(n))^\top \mathrm{kl}(\hat{\theta}(n), \theta) > f(n)\},$$
$$H_{i,n} = \{M_i(n) = 1, |\hat{\theta}_i(n) - \theta_i| \geq m^{-1}\Delta_{\min}/2\}, \quad H_n = \cup_{i=1}^d H_{i,n},$$
$$F_n = \{\Delta_{M(n)} \leq 2h_{T,t(n),M(n)}\}.$$

Then the regret can be bounded as:

$$R^\pi(T) = \mathbb{E}[\sum_{n=1}^T \Delta_{M(n)}] \leq \mathbb{E}[\sum_{n=1}^T \Delta_{M(n)}(\mathbb{1}\{G_n\} + \mathbb{1}\{H_n\})] + \mathbb{E}[\sum_{n=1}^T \Delta_{M(n)}\mathbb{1}\{\overline{G_n}, \overline{H_n}\}]$$
$$\leq m\mathbb{E}[\sum_{n=1}^T (\mathbb{1}\{G_n\} + \mathbb{1}\{H_n\})] + \mathbb{E}[\sum_{n=1}^T \Delta_{M(n)}\mathbb{1}\{\overline{G_n}, \overline{H_n}\}],$$

since $\Delta_{M(n)} \leq m$.

Next we show that for any $n$ such that $M(n) \neq M^\star$, it holds that $\overline{G_n \cup H_n} \subset F_n$. Recall that $c_M(n) \geq b_M(n)$ for any $M$ and $n$ (Theorem 3). Moreover, if $\overline{G_n}$ holds, we have $(M^\star t(n))^\top \mathrm{kl}(\hat{\theta}(n), \theta) \leq f(n)$, which by definition of $b_M$ implies: $b_{M^\star}(n) \geq M^{\star\top}\theta$. Hence we have:

$$
\begin{aligned}
\mathbb{1}\{\overline{G_n}, \overline{H_n}, M(n) \neq M^\star\} &= \mathbb{1}\{\overline{G_n}, \overline{H_n}, \xi_{M(n)}(n) \geq \xi_{M^\star}(n)\} \\
&\leq \mathbb{1}\{\overline{H_n}, c_{M(n)}(n) \geq M^{\star\top}\theta\} \\
&= \mathbb{1}\{\overline{H_n}, M(n)^\top\hat{\theta}(n) + h_{n,t(n),M(n)} \geq M^{\star\top}\theta\} \\
&\leq \mathbb{1}\{M(n)^\top\theta + \Delta_{M(n)}/2 + h_{n,t(n),M(n)} \geq M^{\star\top}\theta\} \\
&= \mathbb{1}\{2h_{n,t(n),M(n)} \geq \Delta_{M(n)}\} \\
&\leq \mathbb{1}\{2h_{T,t(n),M(n)} \geq \Delta_{M(n)}\} \\
&= \mathbb{1}\{F_n\},
\end{aligned}
$$

where the second inequality follows from the fact that event $\overline{G_n}$ implies: $M(n)^\top \hat\theta(n) \leq M(n)^\top \theta + \Delta_{\min}/2 \leq M(n)^\top \theta + \Delta_{M(n)}/2$.

Hence, the regret is upper bounded by:

$$R^\pi(T) \leq m\mathbb{E}[\sum_{n=1}^T \mathbb{1}\{G_n\}] + m\mathbb{E}[\sum_{n=1}^T \mathbb{1}\{H_n\}] + \mathbb{E}[\sum_{n=1}^T \Delta_{M(n)}\mathbb{1}\{F_n\}].$$

We will prove the following inequalities: (i) $\mathbb{E}[\sum_{n=1}^T \mathbb{1}\{G_n\}] \leq m^{-1}C_m'$, with $C_m' \geq 0$ independent of $\theta$, $d$, and $T$, (ii) $\mathbb{E}[\sum_{n=1}^T \mathbb{1}\{H_n\}] \leq 4dm^2\Delta_{\min}^{-2}$, and (iii) $\mathbb{E}[\sum_{n=1}^T \Delta_{M(n)}\mathbb{1}\{F_n\}] \leq 16d\sqrt{m}\Delta_{\min}^{-1}f(T)$.

Hence as announced:

$$R^\pi(T) \leq 16d\sqrt{m}\Delta_{\min}^{-1}f(T) + 4dm^3\Delta_{\min}^{-2} + C_m'.$$

**Inequality (i):** An application of Lemma 1 gives

$$\mathbb{E}[\sum_{n=1}^T \mathbb{1}\{G_n\}] = \sum_{n=1}^T \mathbb{P}[(M^\star t(n))^\top \mathrm{kl}(\hat\theta(n), \theta) > f(n)]$$

$$\leq 1 + \sum_{n\geq 2} C_m n^{-1}(\log(n))^{-2} \equiv m^{-1}C_m' < \infty.$$

**Inequality (ii):** Fix $i$ and $n$. Define $s = \sum_{n'=1}^n \mathbb{1}\{H_{n',i}\}$. Observe that $H_{n',i}$ implies $M_i(n') = 1$, hence $t_i(n) \geq s$. Therefore, applying [4, Lemma B.1], we have that $\sum_{n=1}^T \mathbb{P}[H_{n,i}] \leq 4m^2\Delta_{\min}^{-2}$. Using the union bound: $\sum_{n=1}^T \mathbb{P}[H_n] \leq 4dm^2\Delta_{\min}^{-2}$.

**Inequality (iii):** Let $\ell > 0$. For any $n$ introduce the following events:

$$S_n = \{i \in M(n) : t_i(n) \leq 4mf(T)\Delta_{M(n)}^{-2}\},$$
$$A_n = \{|S_n| \geq \ell\},$$
$$B_n = \{|S_n| < \ell, \ [\exists i \in M(n) : t_i(n) \leq 4\ell f(T)\Delta_{M(n)}^{-2}]\}.$$

We claim that for any $n$ such that $M(n) \neq M^\star$, we have $F_n \subset (A_n \cup B_n)$. To prove this, we show that when $F_n$ holds and $M(n) \neq M^\star$, the event $\overline{A_n \cup B_n}$ cannot happen. Let $n$ be a time instant such that $M(n) \neq M^\star$ and $F_n$ holds, and assume that $\overline{A_n \cup B_n} = \{|S_n| < \ell, \ [\forall i \in M(n) : t_i(n) > 4\ell f(T)\Delta_{M(n)}^{-2}]\}$ happens. Then $F_n$ implies:

$$\Delta_{M(n)} \leq 2h_{T,t(n),M(n)} = 2\sqrt{\frac{f(T)}{2}}\sqrt{\sum_{i\in[d]\setminus S_n}\frac{M_i(n)}{t_i(n)} + \sum_{i\in S_n}\frac{M_i(n)}{t_i(n)}}$$

$$< 2\sqrt{\frac{f(T)}{2}}\sqrt{m\frac{\Delta_{M(n)}^2}{4mf(T)} + |S_n|\frac{\Delta_{M(n)}^2}{4\ell f(T)}} < \Delta_{M(n)}, \tag{9}$$

where the last inequality uses the observation that $\overline{A_n \cup B_n}$ implies $|S_n| < \ell$. Clearly, (9) is a contradiction. Thus $F_n \subset (A_n \cup B_n)$ and consequently:

$$\sum_{n=1}^T \Delta_{M(n)}\mathbb{1}\{F_n\} \leq \sum_{n=1}^T \Delta_{M(n)}\mathbb{1}\{A_n\} + \sum_{n=1}^T \Delta_{M(n)}\mathbb{1}\{B_n\}. \tag{10}$$

To further bound the r.h.s. of the above, we introduce the following events for any $i$:

$$A_{i,n} = A_n \cap \{i \in M(n), \ t_i(n) \leq 4mf(T)\Delta_{M(n)}^{-2}\},$$
$$B_{i,n} = B_n \cap \{i \in M(n), \ t_i(n) \leq 4\ell f(T)\Delta_{M(n)}^{-2}\}.$$

It is noted that:
$$\sum_{i\in[d]} \mathbb{1}\{A_{i,n}\} = \mathbb{1}\{A_n\}\sum_{i\in[d]} \mathbb{1}\{i\in S_n\} = |S_n|\mathbb{1}\{A_n\} \geq \ell\mathbb{1}\{A_n\},$$
and hence: $\mathbb{1}\{A_n\} \leq \frac{1}{\ell}\sum_{i\in[d]}\mathbb{1}\{A_{i,n}\}$. Moreover $\mathbb{1}\{B_n\} \leq \sum_{i\in[d]}\mathbb{1}\{B_{i,n}\}$. Let each basic action $i$ belong to $K_i$ suboptimal arms, ordered based on their gaps as: $\Delta^{i,1} \geq \cdots \geq \Delta^{i,K_i} > 0$. Also define $\Delta^{i,0} = \infty$. Plugging the above inequalities into (10), we have

$$\sum_{n=1}^{T}\Delta_{M(n)}\mathbb{1}\{F_n\} \leq \sum_{n=1}^{T}\sum_{i=1}^{d}\frac{\Delta_{M(n)}}{\ell}\mathbb{1}\{A_{i,n}\} + \sum_{n=1}^{T}\sum_{i=1}^{d}\Delta_{M(n)}\mathbb{1}\{B_{i,n}\}$$

$$= \sum_{n=1}^{T}\sum_{i=1}^{d}\frac{\Delta_{M(n)}}{\ell}\mathbb{1}\{A_{i,n},\ M(n)\neq M^{\star}\} + \sum_{n=1}^{T}\sum_{i=1}^{d}\Delta_{M(n)}\mathbb{1}\{B_{i,n},\ M(n)\neq M^{\star}\}$$

$$\leq \sum_{n=1}^{T}\sum_{i=1}^{d}\sum_{k\in[K_i]}\frac{\Delta^{i,k}}{\ell}\mathbb{1}\{A_{i,n},\ M(n)=k\} + \sum_{n=1}^{T}\sum_{i=1}^{d}\sum_{k\in[K_i]}\Delta^{i,k}\mathbb{1}\{B_{i,n},\ M(n)=k\}$$

$$\leq \sum_{i=1}^{d}\sum_{n=1}^{T}\sum_{k\in[K_i]}\frac{\Delta^{i,k}}{\ell}\mathbb{1}\{i\in M(n),\ t_i(n)\leq 4mf(T)(\Delta^{i,k})^{-2},\ M(n)=k\}$$

$$+ \sum_{i=1}^{d}\sum_{n=1}^{T}\sum_{k\in[K_i]}\Delta^{i,k}\mathbb{1}\{i\in M(n),\ t_i(n)\leq 4\ell f(T)(\Delta^{i,k})^{-2},\ M(n)=k\}$$

$$\leq \frac{8df(T)}{\Delta_{\min}}\left(\frac{m}{\ell}+\ell\right),$$

where the last inequality follows from Lemma 2, which is proven next. The proof is completed by setting $\ell = \sqrt{m}$. $\qquad\square$

**Lemma 2** *Let $C > 0$ be a constant independent of $n$. Then for any $i$ such that $K_i \geq 1$:*

$$\sum_{n=1}^{T}\sum_{k=1}^{K_i}\mathbb{1}\{i\in M(n),\ t_i(n)\leq C(\Delta^{i,k})^{-2},\ M(n)=k\}\Delta^{i,k} \leq \frac{2C}{\Delta_{\min}}.$$

**Proof.** We have:
$$\sum_{n=1}^{T}\sum_{k=1}^{K_i}\mathbb{1}\{i\in M(n),\ t_i(n)\leq C(\Delta^{i,k})^{-2},\ M(n)=k\}\Delta^{i,k}$$

$$= \sum_{n=1}^{T}\sum_{k=1}^{K_i}\sum_{j=1}^{k}\mathbb{1}\{i\in M(n),\ t_i(n)\in(C(\Delta^{i,j-1})^{-2},C(\Delta^{i,j})^{-2}],\ M(n)=k\}\Delta^{i,k}$$

$$\leq \sum_{n=1}^{T}\sum_{k=1}^{K_i}\sum_{j=1}^{k}\mathbb{1}\{i\in M(n),\ t_i(n)\in(C(\Delta^{i,j-1})^{-2},C(\Delta^{i,j})^{-2}],\ M(n)=k\}\Delta^{i,j}$$

$$\leq \sum_{n=1}^{T}\sum_{k=1}^{K_i}\sum_{j=1}^{K_i}\mathbb{1}\{i\in M(n),\ t_i(n)\in(C(\Delta^{i,j-1})^{-2},C(\Delta^{i,j})^{-2}],\ M(n)=k\}\Delta^{i,j}$$

$$\leq \sum_{n=1}^{T}\sum_{j=1}^{K_i}\mathbb{1}\{i\in M(n),\ t_i(n)\in(C(\Delta^{i,j-1})^{-2},C(\Delta^{i,j})^{-2}],\ M(n)\neq M^{\star}\}\Delta^{i,j}$$

$$\leq \frac{C}{\Delta^{i,1}} + \sum_{j=2}^{K_i}C((\Delta^{i,j})^{-2}-(\Delta^{i,j-1})^{-2})\Delta^{i,j}$$

$$\leq \frac{C}{\Delta^{i,1}} + \int_{\Delta^{i,K_i}}^{\Delta^{i,2}}Cx^{-2}\mathrm{d}x \leq \frac{2C}{\Delta^{i,K_i}} \leq \frac{2C}{\Delta_{\min}},$$

which completes the proof. $\qquad\square$

## 2.5 EPOCH-ESCB: An algorithm with lower computational complexity

ESCB with time horizon $T$ has a complexity of $\mathcal{O}(|\mathcal{M}|T)$ as neither $b_M$ nor $c_M$ can be written as $M^\top y$ for some vector $y \in \mathbb{R}^d$. Since $\mathcal{M}$ typically has exponentially many elements, we deduce that ESCB is not computationally efficient. Assuming that the offline (static) combinatorial problem is solvable in $\mathcal{O}(V(\mathcal{M}))$ time, the complexity of CUCB algorithm in [5] and [2] after $T$ rounds is $\mathcal{O}(V(\mathcal{M})T)$. Thus, if the offline problem is efficiently implementable, i.e., $V(\mathcal{M}) = \mathcal{O}(\text{poly}(d))$, CUCB is efficient, whereas ESCB is not. We next propose an extension to ESCB, called EPOCH-ESCB, that attains almost the same regret as ESCB while enjoying much better computational complexity.

EPOCH-ESCB algorithm in epochs of varying lengths. Epoch $k$ comprises rounds $\{N_k, \ldots, N_{k+1} - 1\}$, where $N_{k+1}$ (and thus the length of the $k$-th epoch) is determined at time $n = N_k$. The algorithm simply consists in playing the arm with the maximal index at the beginning of every epoch, and playing the current leader (i.e., the arm with the highest empirical average reward) in the rest of rounds. If the leader is the arm with the maximal index, the length of epoch $k$ will be set twice as long as the previous epoch $k - 1$, i.e., $N_{k+1} = N_k + 2(N_k - N_{k-1})$. Otherwise, it will be set to 1. In contrast to ESCB, EPOCH-ESCB computes the maximal index infrequently, and more precisely (almost) at an exponentially decreasing rate. Thus, one might expect that after $T$ rounds, the maximal index will be computed $\mathcal{O}(\log(T))$ times. The pseudo-code of EPOCH-ESCB is presented in Algorithm 1.

---

**Algorithm 1** EPOCH-ESCB

**Initialization:** Set $k = 1$ and $N_0 = N_1 = 1$.
**for** $n \geq 1$ **do**
   Compute $L(n) \in \arg\max_{M \in \mathcal{M}} M^\top \hat{\theta}(n)$.
   **if** $n = N_k$ **then**
     Select arm $M(n) \in \arg\max_{M \in \mathcal{M}} \xi_M(n)$.
     **if** $M(n) = L(n)$ **then**
       Set $N_{k+1} = N_k + 2(N_k - N_{k-1})$.
     **else**
       Set $N_{k+1} = N_k + 1$.
     **end if**
     Increment $k$.
   **else**
     Select arm $M(n) = L(n)$.
   **end if**
   Observe the rewards, and update $t_i(n)$ and $\hat{\theta}_i(n), \forall i \in M(n)$.
**end for**

---

We assess the performance of EPOCH-ESCB through numerical experiments in the next subsection, and leave the analysis of its regret as a future work. These experiments corroborate our conjecture that he complexity of EPOCH-ESCB after $T$ rounds will be $\mathcal{O}(V(\mathcal{M})T + \log(T)|\mathcal{M}|)$. Compared to CUCB, the complexity is penalized by $|\mathcal{M}|\log(T)$, which may become dominated by the term $V(\mathcal{M})T$ as $T$ grows large.

## 2.6 Numerical Experiments

In this section, we compare the performance of ESCB against existing algorithms through numerical experiments for some classes of $\mathcal{M}$. When implementing ESCB we replace $f(n)$ by $\log(n)$, ignoring the term proportional to $\log(\log(n))$, as is done when implementing KL-UCB in practice.

### 2.6.1 Experiment 1: Matching

In our first experiment, we consider the matching problem with $N_1 = N_2 = 5$, which corresponds to $d = 5^2 = 25$ and $m = 5$. We also set $\theta$ such that $\theta_i = a$ if $i \in M^\star$, and $\theta_i = b$ otherwise, with $0 < b < a < 1$. In this case the lower bound becomes $c(\theta) = \frac{m(m-1)(a-b)}{2\text{kl}(b,a)}$.

Figure 5(a)-(b) depicts the regret of various algorithms for the case of $a = 0.7$ and $b = 0.5$. The curves in Figure 5(a) are shown with a 95% confidence interval. We observe that ESCB-1 has

Figure 5: Regret of various algorithms for matchings with $a = 0.7$ and $b = 0.5$.

Figure 6: Regret of various algorithms for matchings with $a = 0.95$ and $b = 0.3$.

the lowest regret. Moreover, ESCB-2 significantly outperforms CUCB and LLR, and is close to ESCB-1. Moreover, we observe that the regret of EPOCH-ESCBattains is quite close to that of ESCB-2.

Figures 6(a)-(b) presents the regret of various algorithms for the case of $a = 0.95$ and $b = 0.3$. The difference compared to the former case is that ESCB-1 significantly outperforms ESCB-2. The reason is that in the former case, mean rewards of the most of the basic actions were close to 1/2, for which the performance of UCB-type algorithms are closer to their KL-divergence based counterparts. On the other hand, when mean rewards are not close to 1/2, there exists a significant performance gap between ESCB-1 and ESCB-2. Comparing the results with the 'lower bound' curve, we highlight that ESCB-1 gives close-to-optimal performance in both cases. Furthermore, similar to previous experiment, EPOCH-ESCBattains a regret whose curve is almost indistinguishable from that of ESCB-2.

The number of epochs in EPOCH-ESCB vs. time for the two examples is displayed in Figure 7(a)-(b), where the curves are shown with 95% confidence intervals. We observe that in both cases, the number of epochs grows at a rate proportional to $\log(n)/n$ at round $n$. Since the number of epochs is equal to the number of times the algorithm computes indexes, these curves suggest that index computation after $n$ rounds requires a number of operations that scales as $|\mathcal{M}| \log(n)$.

### 2.6.2 Experiment 2: Spanning Trees

In the second experiment, we consider spanning trees problem described in Section 1.4.2 for the case of $N = 5$. In this case, we have $d = \binom{5}{2} = 10$, $m = 4$, and $|\mathcal{M}| = 5^3 = 125$.

Figure 8 portrays the regret of various algorithms with 95% confidence intervals, with $\Delta_{\min} = 0.54$. Our algorithms significantly outperform CUCB and LLR.

(a) $a = 0.7$ and $b = 0.5$          (b) $a = 0.95$, $b = 0.3$

Figure 7: Number of epochs in EPOCH-ESCB vs. time for Experiment 1 and 2 (%95 confidence interval).

(a)          (b)

Figure 8: Regret of various algorithms for spanning trees with $N = 5$ and $\Delta_{\min} = 0.54$.

## 3 Supplementary Material for Adversarial Combinatorial Bandits

### 3.1 Proof of Theorem 6

We first prove a simple result:

**Lemma 3** *For all $x \in \mathbb{R}^d$, we have $\Sigma_{n-1}^+ \Sigma_{n-1} x = \overline{x}$, where $\overline{x}$ is the orthogonal projection of $x$ onto $span(\mathcal{M})$, the linear space spanned by $\mathcal{M}$.*

*Proof:* Note that for all $y \in \mathbb{R}^d$, if $\Sigma_{n-1} y = 0$, then we have

$$y^\top \Sigma_{n-1} y = \mathbb{E}\left[y^\top M M^\top y\right] = \mathbb{E}\left[(y^\top M)^2\right] = 0, \tag{11}$$

where $M$ has law $p_{n-1}$ such that $\sum_M M_i p_{n-1}(M) = q'_{n-1}(i)$, $\forall i \in [d]$ and $q'_{n-1} = (1-\gamma)q_{n-1} + \gamma\mu^0$. By definition of $\mu^0$, each $M \in \mathcal{M}$ has a positive probability. Hence, by (11), $y^\top M = 0$ for all $M \in \mathcal{M}$. In particular, we see that the linear application $\Sigma_{n-1}$ restricted to $span(\mathcal{M})$ is invertible and is zero on $span(\mathcal{M})^\perp$, hence we have $\Sigma_{n-1}^+ \Sigma_{n-1} x = \overline{x}$. □

**Lemma 4** *We have for any $\eta \leq \frac{\gamma\lambda}{m^{3/2}}$ and any $q \in \mathcal{P}$,*

$$\sum_{n=1}^T q^\top \tilde{X}(n) - \sum_{n=1}^T q_{n-1}^\top \tilde{X}(n) \leq \frac{\eta}{2} \sum_{n=1}^T q_{n-1}^\top \tilde{X}^2(n) + \frac{\mathrm{KL}(q, q_0)}{\eta},$$

*where $\tilde{X}^2(n)$ is the vector that is the coordinate-wise square of $\tilde{X}(n)$.*

*Proof:* We have

$$\mathrm{KL}(q, \tilde{q}_n) - \mathrm{KL}(q, q_{n-1}) = \sum_{i \in [d]} q(i) \log \frac{q_{n-1}(i)}{\tilde{q}_n(i)} = -\eta \sum_{i \in [d]} q(i) \tilde{X}_i(n) + \log Z_n,$$

with

$$\log Z_n = \log \sum_{i \in [d]} q_{n-1}(i) \exp\left(\eta \tilde{X}_i(n)\right)$$

$$\leq \log \sum_{i \in [d]} q_{n-1}(i) \left(1 + \eta \tilde{X}_i(n) + \eta^2 \tilde{X}_i^2(n)\right) \tag{12}$$

$$\leq \eta q_{n-1}^\top \tilde{X}(n) + \eta^2 q_{n-1}^\top \tilde{X}^2(n), \tag{13}$$

where we used $\exp(z) \leq 1 + z + z^2$ for all $|z| \leq 1$ in (12) and $\log(1 + z) \leq z$ for all $z > -1$ in (13). Later we verify the condition for the former inequality.

Hence we have

$$\mathrm{KL}(q, \tilde{q}_n) - \mathrm{KL}(q, q_{n-1}) \leq \eta q_{n-1}^\top \tilde{X}(n) - \eta q^\top \tilde{X}(n) + \eta^2 q_{n-1}^\top \tilde{X}^2(n).$$

Generalized Pythagorean inequality (see Theorem 3.1 in [6]) gives

$$\mathrm{KL}(q, q_n) + \mathrm{KL}(q_n, \tilde{q}_n) \leq \mathrm{KL}(q, \tilde{q}_n).$$

Since $\mathrm{KL}(q_n, \tilde{q}_n) \geq 0$, we get

$$\mathrm{KL}(q, q_n) - \mathrm{KL}(q, q_{n-1}) \leq \eta q_{n-1}^\top \tilde{X}(n) - \eta q^\top \tilde{X}(n) + \eta^2 q_{n-1}^\top \tilde{X}^2(n).$$

Finally, summing over $n$ gives

$$\sum_{n=1}^{T} \left(q^\top \tilde{X}(n) - q_{n-1}^\top \tilde{X}(n)\right) \leq \eta \sum_{n=1}^{T} q_{n-1}^\top \tilde{X}^2(n) + \frac{\mathrm{KL}(q, q_0)}{\eta}.$$

To satisfy the condition for the inequality (12), i.e., $\eta |\tilde{X}_i(n)| \leq 1, \ \forall i \in [d]$, we find the upper bound for $\max_{i \in [d]} |\tilde{X}_i(n)|$ as follows:

$$\max_{i \in [d]} |\tilde{X}_i(n)| \leq \|\tilde{X}(n)\|_2$$

$$= \|\Sigma_{n-1}^+ M(n) Y_n\|_2$$

$$\leq m \|\Sigma_{n-1}^+ M(n)\|_2$$

$$\leq m \sqrt{M(n)^\top \Sigma_{n-1}^+ \Sigma_{n-1}^+ M(n)}$$

$$\leq m \|M(n)\|_2 \sqrt{\lambda_{\max}\left(\Sigma_{n-1}^+ \Sigma_{n-1}^+\right)}$$

$$= m^{3/2} \sqrt{\lambda_{\max}\left(\Sigma_{n-1}^+ \Sigma_{n-1}^+\right)}$$

$$= m^{3/2} \lambda_{\max}\left(\Sigma_{n-1}^+\right)$$

$$= \frac{m^{3/2}}{\lambda_{\min}\left(\Sigma_{n-1}\right)},$$

where $\lambda_{\max}(A)$ and $\lambda_{\min}(A)$ respectively denote the maximum and the minimum nonzero eigenvalue of matrix $A$. Note that $\mu^0$ induces uniform distribution over $\mathcal{M}$. Thus by $q'_{n-1} = (1-\gamma)q_{n-1} + \gamma \mu^0$ we see that $p_{n-1}$ is a mixture of uniform distribution and the distribution induced by $q_{n-1}$. Note that, we have:

$$\lambda_{\min}\left(\Sigma_{n-1}\right) = \min_{\|x\|_2=1, x \in span(\mathcal{M})} x^\top \Sigma_{n-1} x.$$

Moreover, we have

$$x^\top \Sigma_{n-1} x = \mathbb{E}\left[x^\top M(n) M(n)^\top x\right] = \mathbb{E}\left[(M(n)^\top x)^2\right] \geq \gamma \mathbb{E}\left[(M^\top x)^2\right],$$

where in the last inequality $M$ has law $\mu^0$. By definition, we have for any $x \in span(\mathcal{M})$ with $\|x\|_2 = 1$,

$$\mathbb{E}\left[(M^\top x)^2\right] \geq \underline{\lambda},$$

so that in the end, we get $\lambda_{\min}(\Sigma_{n-1}) \geq \gamma\underline{\lambda}$, and hence $\eta|\tilde{X}_i(n)| \leq \frac{\eta m^{3/2}}{\gamma\underline{\lambda}}$, $\forall i \in [d]$. Finally, we choose $\eta \leq \frac{\gamma\underline{\lambda}}{m^{3/2}}$ to satisfy the condition for the inequality we used in (12).

$\square$

We have
$$\mathbb{E}_n\left[\tilde{X}(n)\right] = \mathbb{E}_n\left[Y_n\Sigma_{n-1}^+ M(n)\right] = \mathbb{E}_n\left[\Sigma_{n-1}^+ M(n)M(n)^\top X(n)\right] = \Sigma_{n-1}^+\Sigma_{n-1}X(n) = \overline{X(n)},$$

where the last equality follows from Lemma 3 and $\overline{X(n)}$ is the orthogonal projection of $X(n)$ onto $span(\mathcal{M})$. In particular, for any $mq' \in Co(\mathcal{M})$, we have
$$\mathbb{E}_n\left[mq'^\top \tilde{X}(n)\right] = mq'^\top \overline{X(n)} = mq'^\top X(n).$$

Moreover, we have:
$$\begin{aligned}
\mathbb{E}_n\left[q_{n-1}^\top \tilde{X}^2(n)\right] &= \sum_{i\in[d]} q_{n-1}(i)\mathbb{E}_n\left[\tilde{X}_i^2(n)\right]\\
&= \sum_{i\in[d]} \frac{q'_{n-1}(i) - \gamma\mu^0(i)}{1-\gamma}\mathbb{E}_n\left[\tilde{X}_i^2(n)\right]\\
&\leq \frac{1}{m(1-\gamma)}\sum_{i\in[d]} mq'_{n-1}(i)\mathbb{E}_n\left[\tilde{X}_i^2(n)\right]\\
&= \frac{1}{m(1-\gamma)}\mathbb{E}_n\left[\sum_{i\in[d]} \tilde{M}_i(n)\tilde{X}_i^2(n)\right],
\end{aligned}$$

where $\tilde{M}(n)$ is a random arm with the same law as $M(n)$ and independent of $M(n)$. Note that $\tilde{M}_i^2(n) = \tilde{M}_i(n)$, so that we have
$$\begin{aligned}
\mathbb{E}_n\left[\sum_{i\in[d]} \tilde{M}_i(n)\tilde{X}_i^2(n)\right] &= \mathbb{E}_n\left[X(n)^\top M(n)M(n)^\top\Sigma_{n-1}^+\tilde{M}(n)\tilde{M}(n)^\top\Sigma_{n-1}^+ M(n)M(n)^\top X(n)\right]\\
&\leq m^2\mathbb{E}_n[M(n)^\top\Sigma_{n-1}^+ M(n)],
\end{aligned}$$

where we used the bound $M(n)^\top X(n) \leq m$. By [7, Lemma 15], $\mathbb{E}_n[M(n)^\top\Sigma_{n-1}^+ M(n)] \leq d$, so that we have:
$$\mathbb{E}_n\left[q_{n-1}^\top \tilde{X}^2(n)\right] \leq \frac{md}{1-\gamma}.$$

Observe that
$$\begin{aligned}
\mathbb{E}_n\left[q^{\star\top}\tilde{X}(n) - q'^{\top}_{n-1}\tilde{X}(n)\right] &= \mathbb{E}_n\left[q^{\star\top}\tilde{X}(n) - (1-\gamma)q_{n-1}^\top\tilde{X}(n) - \gamma\mu^{0\top}\tilde{X}(n)\right]\\
&= \mathbb{E}_n\left[q^{\star\top}\tilde{X}(n) - q_{n-1}^\top\tilde{X}(n)\right] + \gamma q_{n-1}^\top X(n) - \gamma\mu^{0\top}X(n)\\
&\leq \mathbb{E}_n\left[q^{\star\top}\tilde{X}(n) - q_{n-1}^\top\tilde{X}(n)\right] + \gamma q_{n-1}^\top X(n)\\
&\leq \mathbb{E}_n\left[q^{\star\top}\tilde{X}(n) - q_{n-1}^\top\tilde{X}(n)\right] + \gamma.
\end{aligned}$$

Using Lemma 4 and the above bounds, we get with $mq^\star$ the optimal arm, i.e. $q^\star(i) = \frac{1}{m}$ iff $M_i^\star = 1$,
$$\begin{aligned}
R^{\text{COMBEXP}}(T) &= \mathbb{E}\left[\sum_{n=1}^T mq^{\star\top}\tilde{X}(n) - \sum_{n=1}^T mq'^{\top}_{n-1}\tilde{X}(n)\right]\\
&\leq \mathbb{E}\left[\sum_{n=1}^T mq^{\star\top}\tilde{X}(n) - \sum_{n=1}^T mq_{n-1}^\top\tilde{X}(n)\right] + m\gamma T\\
&\leq \frac{\eta m^2 dT}{1-\gamma} + \frac{m\log\mu_{\min}^{-1}}{\eta} + m\gamma T,
\end{aligned}$$

since

$$\mathrm{KL}(q^\star, q_0) = -\frac{1}{m} \sum_{i \in M^\star} \log m\mu_i^0 \leq \log \mu_{\min}^{-1}.$$

Choosing $\eta = \gamma C$ with $C = \frac{\lambda}{m^{3/2}}$ gives

$$\begin{aligned}
R^{\textsc{CombExp}}(T) &\leq \frac{\gamma C m^2 dT}{1 - \gamma} + \frac{m \log \mu_{\min}^{-1}}{\gamma C} + m\gamma T \\
&= \frac{Cm^2 d + m - m\gamma}{1 - \gamma}\gamma T + \frac{m \log \mu_{\min}^{-1}}{\gamma C} \\
&\leq \frac{(Cm^2 d + m)\gamma T}{1 - \gamma} + \frac{m \log \mu_{\min}^{-1}}{\gamma C}.
\end{aligned}$$

The proof is completed by setting $\gamma = \dfrac{\sqrt{m \log \mu_{\min}^{-1}}}{\sqrt{m \log \mu_{\min}^{-1}} + \sqrt{C(Cm^2 d + m)T}}$. $\qquad\square$

### 3.2 Proof of Proposition 1

We first provide a simple result:

**Lemma 5** *For all probability vectors $q \in \mathbb{R}_{++}^d$, the KL-divergence $z \mapsto \mathrm{KL}(z, q)$ is 1-strongly convex with respect to the $\|\cdot\|_1$ norm.*

**Proof.** To prove the lemma, it suffices to show that for any $x, y \in \mathcal{P}$:

$$(\nabla \mathrm{KL}(x, q) - \nabla \mathrm{KL}(y, q))^\top (x - y) \geq \|x - y\|_1^2.$$

We have

$$\begin{aligned}
(\nabla \mathrm{KL}(x, q) - \nabla \mathrm{KL}(y, q))^\top (x - y) &= \sum_{i \in [d]} \left(1 + \log \frac{x(i)}{q(i)} - 1 - \log \frac{y(i)}{q(i)}\right)(x(i) - y(i)) \\
&= \sum_{i \in [d]} (1 + \log x(i) - 1 - \log y(i))(x(i) - y(i)) \\
&= \left(\nabla \sum_{i \in [d]} x(i) \log x(i) - \nabla \sum_{i \in [d]} y(i) \log y(i)\right)^\top (x - y) \\
&\geq \|x - y\|_1^2,
\end{aligned}$$

where the last inequality follows from strong convexity of the entropy function $z \mapsto \sum_{i \in [d]} z_i \log z_i$ with respect to the $\|\cdot\|_1$ norm [8, Proposition 5.1]. $\qquad\square$

Recall that $u_n = \arg\min_{p \in \mathcal{P}} \mathrm{KL}(p, \tilde{q}_n)$ and that $q_n$ is an $\epsilon_n$-optimal solution for the projection step, that is

$$\mathrm{KL}(u_n, \tilde{q}_n) \geq \mathrm{KL}(q_n, \tilde{q}_n) - \epsilon_n.$$

Using Lemma 5 and [6, Theorem 3.1], we have

$$\mathrm{KL}(q_n, \tilde{q}_n) - \mathrm{KL}(u_n, \tilde{q}_n) \geq (q_n - u_n)^\top \nabla \mathrm{KL}(u_n, \tilde{q}_n) + \frac{1}{2}\|q_n - u_n\|_1^2 \geq \frac{1}{2}\|q_n - u_n\|_1^2,$$

where we used $(q_n - u_n)^\top \nabla \mathrm{KL}(u_n, \tilde{q}_n) \geq 0$ due to first-order optimality condition for $u_n$. Hence $\mathrm{KL}(q_n, \tilde{q}_n) - \mathrm{KL}(u_n, \tilde{q}_n) \leq \epsilon_n$ implies that $\|q_n - u_n\|_\infty \leq \|q_n - u_n\|_1 \leq \sqrt{2\epsilon_n}$.

Consider $q^\star$, the distribution over $\mathcal{P}$ for the optimal arm, i.e. $q^\star(i) = \frac{1}{m}$ iff $M_i^\star = 1$. Recall that from proof of Lemma 4, for $q = q^\star$ we have

$$\mathrm{KL}(q^\star, \tilde{q}_n) - \mathrm{KL}(q^\star, q_{n-1}) \leq \eta q_{n-1}^\top \tilde{X}(n) - \eta {q^\star}^\top \tilde{X}(n) + \eta^2 q_{n-1}^\top \tilde{X}^2(n). \tag{14}$$

Generalized Pythagorean Inequality (see Theorem 3.1 in [6]) gives

$$\mathrm{KL}(q^\star, \tilde{q}_n) \geq \mathrm{KL}(q^\star, u_n) + \mathrm{KL}(u_n, \tilde{q}_n). \tag{15}$$

Let $\underline{q}_n = \min_{i \in M^\star} q_n(i)$. Observe that

$$
\mathrm{KL}(q^\star, u_n) = \sum_{i \in [d]} q^\star(i) \log \frac{q^\star(i)}{u_n(i)} = -\frac{1}{m} \sum_{i \in M^\star} \log m u_n(i)
$$

$$
\geq -\frac{1}{m} \sum_{i \in M^\star} \log m(q_n(i) + \sqrt{2\epsilon_n}) \geq -\frac{1}{m} \sum_{i \in M^\star} \left( \log m q_n(i) + \frac{\sqrt{2\epsilon_n}}{\underline{q}_n} \right)
$$

$$
\geq -\frac{\sqrt{2\epsilon_n}}{\underline{q}_n} - \frac{1}{m} \sum_{i \in M^\star} \log m q_n(i) = -\frac{\sqrt{2\epsilon_n}}{\underline{q}_n} + \mathrm{KL}(q^\star, q_n),
$$

Plugging this into (15), we get

$$
\mathrm{KL}(q^\star, \tilde{q}_n) \geq \mathrm{KL}(q^\star, q_n) - \frac{\sqrt{2\epsilon_n}}{\underline{q}_n} + \mathrm{KL}(u_n, \tilde{q}_n) \geq \mathrm{KL}(q^\star, q_n) - \frac{\sqrt{2\epsilon_n}}{\underline{q}_n}.
$$

Putting this together with (14) yields

$$
\mathrm{KL}(q^\star, q_n) - \mathrm{KL}(q^\star, q_{n-1}) \leq \eta q_{n-1}^\top \tilde{X}(n) - \eta {q^\star}^\top \tilde{X}(n) + \eta^2 q_{n-1}^\top \tilde{X}^2(n) + \frac{\sqrt{2\epsilon_n}}{\underline{q}_n}.
$$

Finally, summing over $n$ gives

$$
\sum_{n=1}^{T} \left( {q^\star}^\top \tilde{X}(n) - q_{n-1}^\top \tilde{X}(n) \right) \leq \eta \sum_{n=1}^{T} q_{n-1}^\top \tilde{X}^2(n) + \frac{\mathrm{KL}(q^\star, q_0)}{\eta} + \frac{1}{\eta} \sum_{n=1}^{T} \frac{\sqrt{2\epsilon_n}}{\underline{q}_n}.
$$

Defining

$$
\epsilon_n = \frac{\left( \underline{q}_n \log \mu_{\min}^{-1} \right)^2}{32 n^2 \log^3(n+1)}, \quad \forall n \geq 1,
$$

and recalling that $\mathrm{KL}(q^\star, q_0) \leq \log \mu_{\min}^{-1}$, we get

$$
\sum_{n=1}^{T} \left( {q^\star}^\top \tilde{X}(n) - q_{n-1}^\top \tilde{X}(n) \right) \leq \eta \sum_{n=1}^{T} q_{n-1}^\top \tilde{X}^2(n) + \frac{\log \mu_{\min}^{-1}}{\eta} + \frac{\log \mu_{\min}^{-1}}{\eta} \sum_{n=1}^{T} \sqrt{\frac{2}{32 n^2 \log^3(n+1)}}
$$

$$
\leq \eta \sum_{n=1}^{T} q_{n-1}^\top \tilde{X}^2(n) + \frac{2 \log \mu_{\min}^{-1}}{\eta},
$$

where we used the fact $\sum_{n \geq 1} n^{-1} (\log(n+1))^{-3/2} \leq 4$. We remark that by the properties of KL divergence and since $q'_{n-1} \geq \gamma \mu^0 > 0$, we have $\underline{q}_n > 0$ at every round $n$, so that $\epsilon_n > 0$ at every round $n$.

Using the above result and following the same lines as in the proof of Theorem 6, we have

$$
R^{\textsc{CombExp}}(T) \leq \frac{\eta m^2 dT}{1 - \gamma} + \frac{2m \log \mu_{\min}^{-1}}{\eta} + m\gamma T.
$$

Choosing $\eta = \gamma C$ with $C = \frac{\lambda}{m^{3/2}}$ gives

$$
R^{\textsc{CombExp}}(T) \leq \frac{(Cm^2 d + m)\gamma T}{1 - \gamma} + \frac{2m \log \mu_{\min}^{-1}}{\gamma C}.
$$

The proof is completed by setting $\gamma = \dfrac{\sqrt{2m \log \mu_{\min}^{-1}}}{\sqrt{2m \log \mu_{\min}^{-1}} + \sqrt{C(Cm^2 d + m)T}}$. $\qquad\square$

### 3.3 Proof of Theorem 7

We calculate the time complexity of the various steps of COMBEXP at round $n \geq 1$.

(i) Mixing: This step requires $\mathcal{O}(d)$ time.

(ii) Decomposition: Using the algorithm of [9], the vector $mq'_{n-1}$ may be represented as a convex combination of at most $d+1$ arms in $\mathcal{O}(d^4)$ time, so that $p_{n-1}$ may have at most $d+1$ non-zero elements (observe that the existence of such a representation follows from Carathéodory Theorem).

(iii) Sampling: This step takes $\mathcal{O}(d)$ time since $p_{n-1}$ has at most $d+1$ non-zero elements.

(iv) Estimation: The construction of matrix $\Sigma_{n-1}$ is done in time $\mathcal{O}(d^2)$ since $p_n$ has at most $d+1$ non-zero elements and $MM^\top$ is formed in $\mathcal{O}(d)$ time. Computing the pseudo-inverse of $\Sigma_{n-1}$ costs $\mathcal{O}(d^3)$.

(v) Update: This step requires $\mathcal{O}(d)$ time.

(vi) Projection: The projection step is equivalent to solving a convex program up to accuracy $\epsilon_n = \mathcal{O}(n^{-2} \log^{-3}(n))$. We use the Interior-Point Method (Barrier method). The total number of Newton iterations to achieve accuracy $\epsilon_n$ is $\mathcal{O}(\sqrt{s} \log(s/\epsilon_n))$ [10, Ch. 11]. Moreover, the cost of each iteration is $\mathcal{O}((d+c)^3)$ [10, Ch. 10], so that the total cost of this step becomes $\mathcal{O}(\sqrt{s}(c+d)^3 \log(s/\epsilon_n))$. Plugging $\epsilon_n = \mathcal{O}(n^{-2} \log^{-3}(n))$ and noting that $\mathcal{O}(\sum_{n=1}^{T} \log(s/\epsilon_n)) = \mathcal{O}(T \log(T))$, the cost of this step is $\mathcal{O}(\sqrt{s}(c+d)^3 T \log(T))$.

Hence the total time complexity after $T$ rounds is $\mathcal{O}(T[\sqrt{s}(c+d)^3 \log(T) + d^4])$, which completes the proof. $\qquad\square$

### 3.4 Implementation: The Case of Graph Coloring

In this subsection, we present an iterative algorithm for the projection step of COMBEXP, for the graph coloring problem described next.

Consider a graph $G = (V, E)$ consisting of $m$ nodes indexed by $i \in [m]$. Each node can use one of the $c \geq m$ available colors indexed by $j \in [c]$. A feasible coloring is represented by a matrix $M \in \{0, 1\}^{m \times c}$, where $M_{ij} = 1$ if and only if node $i$ is assigned color $j$. Coloring $M$ is feasible if (i) for all $i$, node $i$ uses at most one color, i.e., $\sum_{j \in [c]} M_{ij} \in \{0, 1\}$; (ii) neighboring nodes are assigned different colors, i.e., for all $i, i' \in [m]$, $(i, i') \in E$ implies for all $j \in [c]$, $M_{ij} M_{i'j} = 0$. In the following we denote by $\mathcal{K} = \{\mathcal{K}_\ell, \ell \in [k]\}$ the set of maximal cliques of the graph $G$. We also introduce $K_{\ell i} \in \{0, 1\}$ such that $K_{\ell i} = 1$ if and only if node $i$ belongs to the maximal clique $\mathcal{K}_\ell$.

There is a specific case where our algorithm can be efficiently implementable: when the convex hull $Co(\mathcal{M})$ can be captured by polynomial in $m$ many constraints. Note that this cannot be ensured unless restrictive assumptions are made on the graph $G$ since there are up to $3^{m/3}$ maximal cliques in a graph with $m$ vertices [11]. There are families of graphs in which the number of cliques is polynomially bounded. These families include chordal graphs, complete graphs, triangle-free graphs, interval graphs, and planar graphs. Note however, that a limited number of cliques does not ensure a priori that $Co(\mathcal{M})$ can be captured by a limited number of constraints. To the best of our knowledge, this problem is open and only particular cases have been solved as for the stable set polytope (corresponding to the case $c = 2$, $X_{i1} = 1$ and $X_{i2} = 0$ with our notation) [12].

For the coloring problem described above we have

$$Co(\mathcal{M}) = Co\{\forall i, \sum_{j \in [c]} M_{ij} \leq 1, \quad \forall \ell, j, \sum_{i \in [m]} K_{\ell i} M_{ij} \leq 1\}. \tag{16}$$

Note that in the special case where $G$ is the complete graph, such a representation becomes

$$Co(\mathcal{M}) = Co\{\sum_{j \in [c]} M_{ij} \leq 1, \quad \forall i, \sum_{i \in [m]} M_{ij} \leq 1, \quad \forall j\}.$$

We now give an algorithm for the projection a distribution $p$ onto $\mathcal{P}$ using KL divergence. Since $\mathcal{P}$ is a scaled version of $Co(\mathcal{M})$, we give an algorithm for the projection of $mp$ onto $Co(\mathcal{M})$ given by (16).

Set $\lambda_i(0) = \mu_j(0) = 0$ for all $i, j$ and then define for $t \geq 0$,

$$\forall i \in [m], \; \lambda_i(t+1) = \log\Big(\sum_j mp_{ij} e^{-\mu_j(t)}\Big) \tag{17}$$

$$\forall j \in [c], \; \mu_j(t+1) = \max_\ell \log\Big(\sum_i K_{i\ell} mp_{ij} e^{-\lambda_i(t+1)}\Big). \tag{18}$$

We can show that

**Proposition 1** *Let $p_{ij}^\star = \lim_{t\to\infty} p_{ij} e^{-\lambda_i(t)-\mu_j(t)}$. Then $mp^\star$ is the projection of $mp$ onto $Co(\mathcal{M})$ using the KL divergence.*

Although this algorithm is shown to converge, we must stress that the step (18) might be expensive as the number of distinct values of $\ell$ might be exponential in $m$. When $G$ is a complete graph, this step is easy and our algorithm reduces to Sinkhorn's algorithm (see [13] for a discussion).

*Proof:* First note that the definition of projection can be extended to non-negative vectors thanks to the relation

$$\mathrm{KL}(p^\star, q) = \min_{p \in \Xi} \mathrm{KL}(p, q).$$

More precisely, given an alphabet $A$ and a vector $q \in \mathbb{R}_+^A$, we have for any probability vector $p \in \mathbb{R}_+^A$

$$\sum_{a \in A} p(a) \log \frac{p(a)}{q(a)} \geq \sum_a p(a) \log \frac{\sum_a p(a)}{\sum_a q(a)} = \log \frac{1}{\|q\|_1},$$

thanks to the log-sum inequality. Hence we see that $p^\star(a) = \frac{q(a)}{\|q\|_1}$ is the projection of $q$ onto the simplex of $\mathbb{R}_+^A$.

Now define $\mathcal{A}_i = Co\{M_{ij}, \sum_j M_{ij} \leq 1\}$ and $\mathcal{B}_{\ell j} = Co\{M_{ij}, \sum_i K_{\ell i} M_{ij} \leq 1\}$. Hence $\bigcap_i \mathcal{A}_i \bigcap \bigcap_{\ell j} \mathcal{B}_{\ell j} = Co(\mathcal{M})$. By the argument described above, iteration (17) (resp. (18)) corresponds to the projection onto $\mathcal{A}_i$ (resp. $\bigcap_\ell \mathcal{B}_{\ell j}$) and the proposition follows from Theorem 5.1 in [6]. $\qquad\square$

### 3.5 Examples

In this subsection, we compare the performance of COMBEXP against state-of-the-art algorithms (refer to Table 2 for the summary of regret of various algorithms).

#### 3.5.1 $m$-sets

In this case, $\mathcal{M}$ is the set of all $d$-dimensional binary vectors with $m$ ones. We have

$$\mu_{\min} = \min_i \frac{1}{\binom{d}{m}} \sum_M M_i = \frac{\binom{d-1}{m-1}}{\binom{d}{m}} = \frac{m}{d}.$$

Moreover, according to [7, Proposition 12], we have $\underline{\lambda} = \frac{m(d-m)}{d(d-1)}$. When $m = o(d)$, the regret of COMBEXP becomes $O(\sqrt{m^3 dT \log(d/m)})$, namely it has the same performance as COMBAND and EXP2 WITH JOHN'S EXPLORATION.

#### 3.5.2 Matching

Let $\mathcal{M}$ be the set of perfect matchings in $\mathcal{K}_{m,m}$, where we have $d = m^2$ and $|\mathcal{M}| = m!$. We have

$$\mu_{\min} = \min_i \frac{1}{m!} \sum_M M_i = \frac{(m-1)!}{m!} = \frac{1}{m},$$

Furthermore, from [7, Proposition 4] we have that $\underline{\lambda} = \frac{1}{m-1}$, thus giving $R^{\text{COMBEXP}}(T) = O(\sqrt{m^5 T \log(m)})$, which is the same as the regret of COMBAND and EXP2 WITH JOHN'S EXPLORATION in this case.

### 3.5.3 Spanning Trees

In our next example, we assume that $\mathcal{M}$ is the set of spanning trees in the complete graph $\mathcal{K}_N$. In this case, we have $d = \binom{N}{2}$, $m = N - 1$, and by Cayley's formula $\mathcal{M}$ has $N^{N-2}$ elements. Observe that

$$\mu_{\min} = \min_i \frac{1}{N^{N-2}} \sum_M M_i = \frac{(N-1)^{N-3}}{N^{N-2}},$$

which gives for $N \geq 2$

$$\log \mu_{\min}^{-1} = \log \left( \frac{N^{N-2}}{(N-1)^{N-3}} \right)$$

$$= (N-3) \log \left( \frac{N}{N-1} \right) + \log N$$

$$\leq (N-3) \log 2 + \log(N) \leq 2N.$$

From [7, Corollary 7], we also get $\underline{\lambda} \geq \frac{1}{N} - \frac{17}{4N^2}$. For $N \geq 6$, the regret of COMBAND takes the form $O(\sqrt{N^5 T \log(N)})$ since $\frac{m}{d\underline{\lambda}} < 7$ when $N \geq 6$. Further, EXP2 WITH JOHN'S EXPLORATION attains the same regret. On the other hand, we get

$$R^{\text{COMBEXP}}(T) = O(\sqrt{N^5 T \log(N)}), \quad N \geq 6,$$

and therefore it gives the same regret as COMBAND and EXP2 WITH JOHN'S EXPLORATION.

### 3.5.4 Cut sets

Consider the case where $\mathcal{M}$ is the set of balanced cuts of the complete graph $\mathcal{K}_{2N}$, where a balanced cut is defined as the set of edges between a set of $N$ vertices and its complement. It is easy to verify that $d = \binom{2N}{2}$ and $m = N^2$. Moreover, $\mathcal{M}$ has $\binom{2N}{N}$ balanced cuts and hence

$$\mu_{\min} = \min_i \frac{1}{\binom{2N}{N}} \sum_M M_i = \frac{\binom{2N-2}{N-1}}{\binom{2N}{N}} = \frac{N}{4N-2},$$

Moreover, by [7, Proposition 9], we have

$$\underline{\lambda} = \frac{1}{4} + \frac{8N-7}{4(2N-1)(2N-3)}, \quad N \geq 2,$$

and consequently, the regret of COMBEXP becomes $O(N^4 \sqrt{T})$ for $N \geq 2$, which is the same as that of COMBAND and EXP2 WITH JOHN'S EXPLORATION.