[Reviews · NeurIPS 2015]

Submitted by Assigned_Reviewer_1

The authors derive regret lower bounds for stochastic combinatorial bandit problems with semi-bandit feedback, and improved algorithms for stochastic and adversarial bandit problems. The proposed ESCB algorithm improves performance guarantees in the stochastic setting, and the proposed COMBEXP algorithm matches the regret of existing algorithms but can do so with decreased computational complexity.

Quality: The theoretical results appear quite strong. The authors mention that their ESCB algorithm empirically outperforms existing algorithms, but no empirical data or experiments are provided.

Clarity: The introduction is quite clear and well-written, as is Section 3. It would be helpful if the authors provide a bit more of an intuitive description of the proposed algorithms, especially for Eq 2 and line 264.

Summary: The paper is well-written and provides a number of theoretical and algorithmic advances for combinatorial bandit problems in both stochastic and adversarial conditions. While the writing may be suitable for experts in the field, it could benefit from more high-level descriptions of underlying intuition throughout the paper.

Submitted by Assigned_Reviewer_2

Theorem 1 and the intuition behind (which is used to develop Algorithm 1) is the most interesting part of the paper. It reminded me of this paper http://arxiv.org/abs/1405.4758 which should be cited. The contribution on the adversarial case are too incremental in my opinion, but the result for the stochastic case could be enough to justify publication in NIPS.
Summary: Some limited advances on combinatorial bandits. The paper is definitely above the average paper on this topic and pass the bar for NIPS.

Submitted by Assigned_Reviewer_3

The contributions of this paper are threefold:

1) For the stochastic combinatorial semi-bandits, the authors have derived a problem-specific lower bound and discussed its scaling.

2) For the stochastic combinatorial semi-bandits, the authors have achieved an improved regret bound under some *additional assumptions*.

3) For the adversarial combinatorial bandits, the authors have proposed an algorithm with the same regret scaling as the state of the arts, but with lower computational complexity for some problems.

To the best of my knowledge, all these contributions are novel and nontrivial, though I have not seen a *fundamentally new* idea in this paper. The authors are quite familiar with the literature and have done a good literature review. I think the paper is ready to be published in NIPS after some appropriate revisions.

My major comment is that the authors should position the paper better with respect to some existing literature. In particular, with respect to Kveton et al. 2015 [11], contribution 2) of this paper has achieved better regret bound under some *additional assumptions* (i.e. this paper assumes the rewards are Bernoulli and independent across actions, while [11] does not make these assumptions). The authors should clarify these differences in the next version of this paper. Moreover, since [11] claims that their regret bound is tight, to avoid confusion, the authors should explicitly mention that the particular example [11] used to derive their lower bound does not satisfy the assumptions of this paper.

There are some recent papers on combinatorial bandits/semi-bandits, and the authors might consider citing them:

http://jmlr.org/proceedings/papers/v40/Neu15.html http://jmlr.org/proceedings/papers/v37/wen15

Some minor comments:

1) Line 222, \Delta^M should be \Delta_M;

2) In equation 2, both M and t(n) are vectors, so Mt(n) is not well defined.
Summary: To the best of my knowledge, the contributions of this paper are novel and nontrivial, though I have not seen a *fundamentally new* idea in this paper. I think the paper is ready to be published in NIPS after some appropriate revisions.

Author Feedback
Author rebuttal: We would like to thank the reviewers for their constructive comments and kind words. We provide point-to-point answers to the concerns raised by the reviewers.

Assigned_Reviewer_1: - Due to lack of space, numerical experiments are presented in the supplementary material.
- The intuition behind our algorithms is the following. Our algorithms are optimistic: we calculate high probability upper bounds for rewards of each decision. Our algorithms are based on a recent concentration inequality of sums of KL-divergences (Lemma 1 in the supplement), roughly stating that ( M t(n) )^T kl(\hat \theta(n) , \theta) \leq log(n) with high probability. So to find an high probability upper bound on M^T \theta, it is sufficient to maximize M^T q, under the constraint that ( M t(n) )^T kl(\hat \theta(n) , q) \leq log(n). This is precisely the definition of b_M.

Assigned_Reviewer_3: - The contribution with respect to Kveton et al. 2015 will be made precise. As the reviewer wrote, we use the additional assumption that, for all n, X_1(n), ... , X_d(n) are independent. The lower bound derived by Kveton et al. does not satisfy our assumptions, so that there is no contradiction between our work and theirs.
- We will add references to the papers of Wen and Neu as suggested since they are relevant.
- Indeed \Delta^M should be \Delta_M
- We omitted to state the convention that, given vectors x = (x_i) ,y = (y_i) , we denote by x y = (x_i y_i)_i their component-by-component product.

Assigned_Reviewer_4: - We cited the conference version of http://arxiv.org/abs/1405.4758 (reference [21], COLT 2014).

Assigned_Reviewer_5: We assume that set {\cal M} is what the reviewer calls "concept class". We analyzed the complexity of the projection step for particular cases such as the set of matchings. More generally, we considered the case where the convex hull of {\cal M} is described by polynomially many linear inequalities. To the best of our knowledge, if {\cal M} is a general set it seems highly non-trivial to calculate the complexity of the projection step.

Assigned_Reviewer_6: As discussed in section 4.2, in the stochastic case, our results still hold if we do not assume that ||M||_1 = m for all M. We simply require || M ||_1 \leq m for all M for our results to hold. In the adversarial case, the assumption ||M||_1 = m for all M is frequently used in the literature, see for instance [12].